# Design principles of autocatalytic cycles constrain enzyme kinetics and force low substrate saturation at flux branch points

Uri Barenholz[1]*, Dan Davidi[1], Ed Reznik[2,3], Yinon Bar-On[1], Niv Antonovsky[1], Elad Noor[4], Ron Milo[1]*

[1]Department of Plant and Environmental Sciences, The Weizmann Institute of Science, Rehovot, Israel; [2]Computational Biology Program, Memorial Sloan Kettering Cancer Center, New York, United States; [3]Center for Molecular Oncology, Memorial Sloan Kettering Cancer Center, New York, United States; [4]Institute of Molecular Systems Biology, ETH Zurich, Zurich, Switzerland

**Abstract** A set of chemical reactions that require a metabolite to synthesize more of that metabolite is an autocatalytic cycle. Here, we show that most of the reactions in the core of central carbon metabolism are part of compact autocatalytic cycles. Such metabolic designs must meet specific conditions to support stable fluxes, hence avoiding depletion of intermediate metabolites. As such, they are subjected to constraints that may seem counter-intuitive: the enzymes of branch reactions out of the cycle must be overexpressed and the affinity of these enzymes to their substrates must be relatively weak. We use recent quantitative proteomics and fluxomics measurements to show that the above conditions hold for functioning cycles in central carbon metabolism of *E. coli*. This work demonstrates that the topology of a metabolic network can shape kinetic parameters of enzymes and lead to seemingly wasteful enzyme usage.

**\*For correspondence:** uri. barenholz@gmail.com (UB); ron. milo@weizmann.ac.il (RM)

**Competing interests:** The authors declare that no competing interests exist.

## Introduction

An essential trait of living systems is their ability to reproduce. This fundamental ability makes all living organisms autocatalytic by definition. Moreover, autocatalytic metabolism is considered to be one of the essential components of life (*Ganti et al., 2003*).

In this work, we focus on autocatalytic cycles in chemical reaction systems, in the context of metabolic networks. The components we consider are the metabolites of the system, with autocatalytic cycles being formed using the reactions of the metabolic network. An illustrative example for a metabolic autocatalytic cycle is glycolysis. In glycolysis, 2 ATP molecules are consumed in the priming phase, in order to produce 4 ATP molecules in the pay off phase. Therefore, in order to produce ATP in glycolysis, ATP must already be present in the cell. Subsequently, autocatalysis of ATP in glycolysis (also referred to as 'turbo design') results in sensitivity to mutations in seemingly irrelevant enzymes (*Teusink et al., 1998*). Autocatalytic cycles have also been shown to be optimal network topologies that minimize the number of reactions needed for the production of precursor molecules from different nutrient sources (*Riehl et al., 2010*).

Metabolic networks often require the availability of certain intermediate metabolites, in addition to the nutrients consumed, in order to function. Examples of obligatorily autocatalytic internal metabolites in different organisms, on top of ATP, are NADH, and coenzyme A (*Kun et al., 2008*). We find that other central metabolites, such as phospho-sugars and organic acids, are autocatalytic under common growth conditions. The requirement for availability of certain metabolites in order to consume nutrients implies metabolic processes must be finely controlled to prevent such essential

**eLife digest** Many bacteria are able to produce all the molecules they need to survive from a limited supply of nutrients. This allows the bacteria to thrive even in harsh environments where other organisms struggle to live. The bacteria act as miniature chemical factories to convert nutrients into the desired molecules via a series of chemical reactions. Some molecules are made in sets of reactions termed autocatalytic cycles. These reaction sets require a molecule to be present in the cell in order to produce more of that molecule; like how a savings account needs to contain some money before it can generate more via interest.

Bacteria have many different enzymes that each drive specific chemical reactions. In order for an autocatalytic cycle to work properly, the cell needs to maintain adequate supplies of the molecule it is trying to make and all of the "intermediate" molecules in the cycle. If less of an intermediate molecule is produced, for example, the cell needs to reduce the demand for that molecule by controlling later chemical reactions in the cycle. Bacteria control chemical reactions by regulating the activities of the enzymes involved, but it is not clear exactly how they regulate the enzymes that drive autocatalytic cycles.

Barenholz et al. combined two approaches called proteomics and fluxomics to study autocatalytic cycles in a bacterium known as *E. coli*. The experiments suggest several core principles allow autocatalytic cycles to work smoothly in the bacteria. The next step is to apply these principles to different kinds of molecules produced in bacterial cells. A future challenge is to search for other structures that regulate chemical reactions in *E. coli* and other bacteria. Extending our understanding of autocatalytic cycles and other pathways of chemical reactions is essential for designing and engineering new reactions in bacteria. Such knowledge can be used to modify bacteria to produce valuable chemicals in environmentally friendly ways.

metabolites from running out; in such cases metabolism will come to a halt. Autocatalytic cycles present control challenges because the inherent feed-back nature of autocatalytic cycles makes them susceptible to instabilities such as divergence or drainage of their intermediate metabolites (*Teusink et al., 1998*; *Fell et al., 1999*; *Reznik and Segrè, 2010*). The stability criteria typically represent one constraint among the parameters of the cycle enzymes. For large cycles, such as the whole metabolic network, one such constraint adds little information. For compact autocatalytic cycles embedded within metabolism, one such constraint is much more informative. We thus focus our efforts on analyzing small autocatalytic cycles. Finding the unique constraints that metabolic autocatalytic cycles impose is essential for understanding the limitations of existing metabolic networks, as well as for modifying them for synthetic biology and metabolic engineering applications.

A key example of an autocatalytic cycle in carbon metabolism is the Calvin-Benson-Bassham cycle (CBB) (*Benson et al., 1950*). The carbon fixation CBB cycle, which fixes $CO_2$ while transforming five-carbon compounds into two three-carbon compounds, serves as the main gateway for converting inorganic carbon to organic compounds in nature (*Raven et al., 2012*). The autocatalytic nature of the CBB cycle stems from the fact that for every 5 five-carbon compounds the cycle consumes, 6 five-carbon compounds are produced (by the fixation of 5 $CO_2$ molecules). Beyond the CBB cycle, we show that most of the reactions and metabolites in the core of central carbon metabolism are part of compact (i.e. consisting of around 10 reactions or fewer) metabolic autocatalytic cycles. Some of the autocatalytic cycles we find are not usually considered as such. The span of autocatalytic cycles in central carbon metabolism suggests that the constraints underlying their stable operation have network-wide biological consequences.

In this study, we present the specific requirements metabolic autocatalytic cycles must meet in order to achieve at least one, non-zero, steady state which is stable in respect to fluctuations of either metabolites or enzyme levels close to the steady state point. The mathematical tools we use are part of dynamical systems theory (*Strogatz, 2014*). We identify the kinetic parameters of enzymes at metabolic branch points out of an autocatalytic cycle as critical values that determine whether the cycle can operate stably. We show that the affinity of enzymes consuming intermediate metabolites of autocatalytic cycles must be limited to prevent depletion of these metabolites.

Moreover, we show that the stable operation of such cycles requires low saturation, and thus excess expression, of these enzymes. Low saturation of enzymes has previously been suggested to stem from a number of reasons in different contexts: (A) to achieve a desired flux in reactions close to equilibrium, for example in glycolysis (*Staples and Suarez, 1997*; *Eanes et al., 2006*; *Flamholz et al., 2013*); (B) to provide safety factors in the face of varying nutrient availability, for example in the brush-border of the mouse intestine (*Weiss et al., 1998*); (C) to accommodate rapid shifts in demand from the metabolic networks in muscles with low glycolytic flux (*Suarez et al., 1997*); (D) to allow fast response times, for example to pulses of oxidative load in erythrocytes, resulting from their adherence to phagocytes (*Salvador and Savageau, 2003*). Our findings add to these reasons the essential stabilizing effect of low saturation of branch reactions on the stability of fluxes through autocatalytic cycles.

We use recent fluxomics and proteomics data to test the predictions we make. We find them to hold in all cases tested where autocatalytic cycles support flux. Our analysis demonstrates how the requirement for stable operation of autocatalytic cycles results in design principles that are followed by autocatalytic cycles in-vivo. The results and design principles presented here can be further used in synthetic metabolic engineering applications that require proper functioning of autocatalytic cycles.

## Results

### Compact autocatalytic cycles are common and play important roles in the core of central carbon metabolism

Different definitions exist for autocatalytic sets in the context of chemical reaction networks (*Hordijk and Steel, 2004*; *Eigen and Schuster, 2012*; *Kun et al., 2008*). Here we define an autocatalytic cycle as a set of reactions and metabolites that form a cycle, and that, when the reactions are applied to the substrates at the given stoichiometric ratios, increase the amount of the intermediate metabolites. A minimal example of a metabolic autocatalytic cycle is shown in *Figure 1*, where an internal metabolite joins with an external assimilated metabolite to give rise to $1 + \delta$ copies of the internal metabolite, representing an increase by $\delta$ copies. For stable operation, $\delta$ copies have to branch out of the cycle, and this consumption must be robust to small fluctuations in enzyme levels

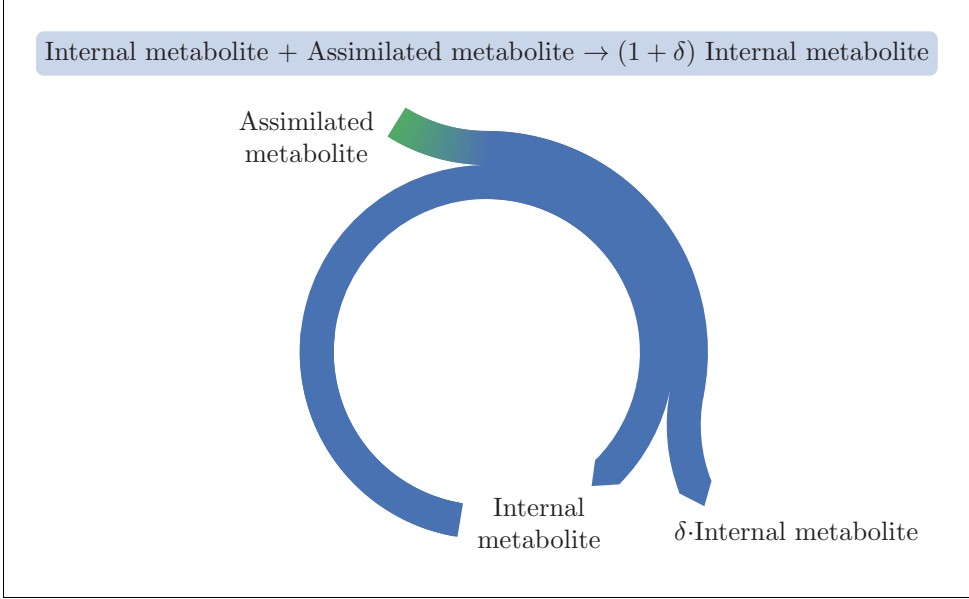

**Figure 1.** A basic autocatalytic cycle requires an internal metabolite to be present in order to assimilate the external metabolite into the cycle, increasing the amount of the internal metabolite by some amount, $\delta$.

and metabolite concentrations. For a formal, mathematical definition, see Materials and methods section "Formal definition of an autocatalytic metabolic cycle".

While rarely discussed as such, a systematic search in the central carbon metabolism core model of *E. coli* (see Materials and methods section "Systematic identification of autocatalytic cycles in metabolic networks") shows the ubiquity of compact autocatalytic cycles. On top of the previously discussed CBB cycle (*Figure 2*, example I), we show two other prominent examples:

- The glyoxylate cycle within the TCA cycle, which turns an internal malate and two external acetyl-CoAs into two malate molecules. This is achieved by transforming malate to isocitrate, while assimilating acetyl-CoA, and then splitting the isocitrate to produce two malate molecules, assimilating another acetyl-CoA (*Kornberg, 1966*) (*Figure 2*, example II).

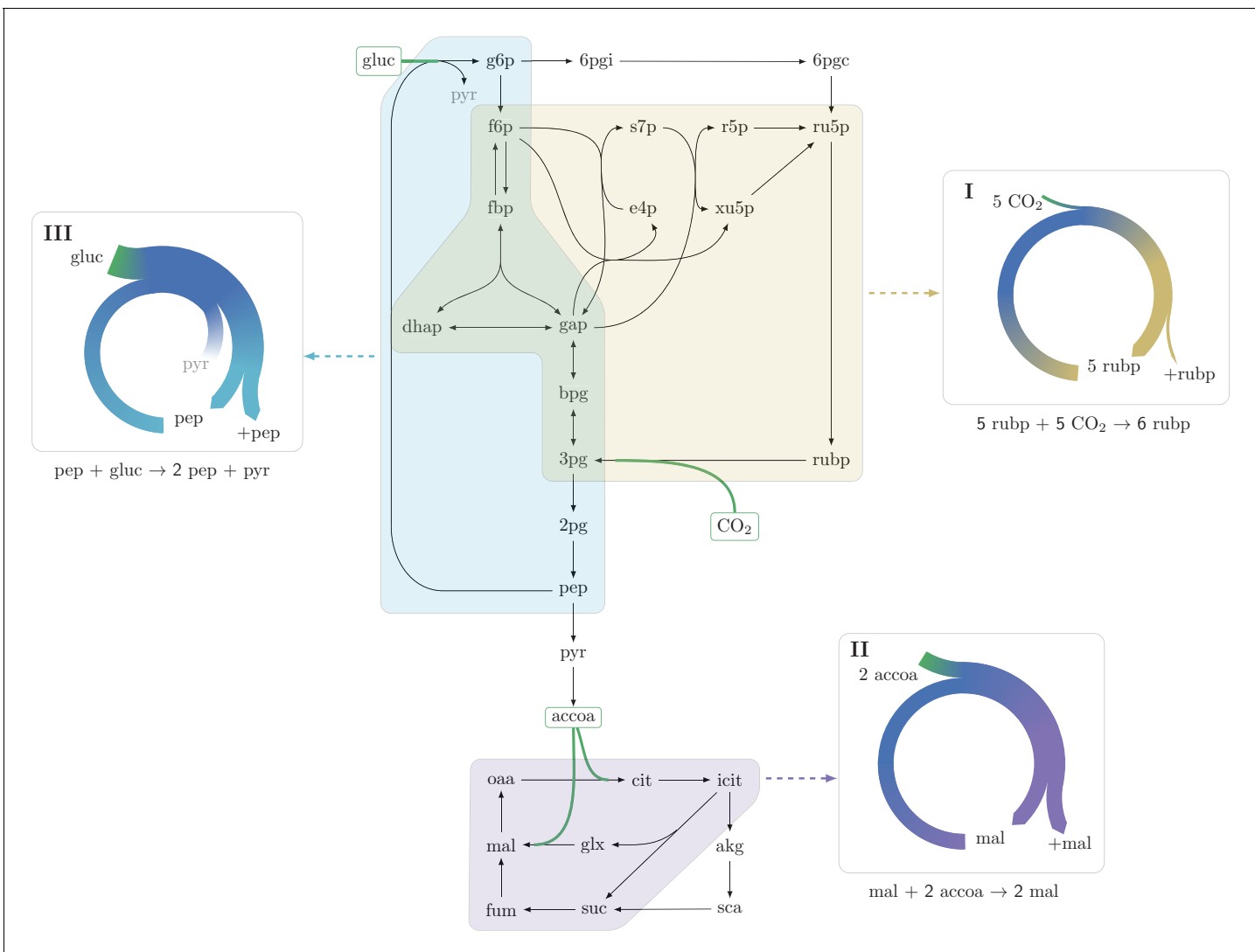

**Figure 2.** Three representative autocatalytic cycles in central carbon metabolism: (I) The Calvin-Benson-Bassham cycle (yellow); (II) The glyoxylate cycle (magenta); (III) A cycle using the phosphotransferase system (PTS) to assimilate glucose (cyan). Assimilation reactions are indicated in green. Arrow width in panels represent the relative carbon flux.

The following figure supplements are available for figure 2:

**Figure supplement 1.** An autocatalytic cycle assimilating ribose-5-phosphate using the pentose phosphate pathway.

**Figure supplement 2.** An autocatalytic cycle assimilating dhap while consuming gap using the fba reaction in the gluconeogenic direction.

- A cycle formed by the glucose phosphotransferase system (PTS) in bacteria. This transport system imports a glucose molecule using phosphoenolpyruvate (pep) as a co-factor. The imported glucose is further catabolized, producing two pep molecules via glycolysis (*Figure 2*, example III).

Two additional examples are presented in *Figure 2—figure supplements 1* and *2* and discussed below.

The ubiquity of compact autocatalytic cycles in the core of central carbon metabolism motivates the study of unique features of autocatalytic cycles, as derived below, which may constrain and shape the kinetic parameters of a broad set of enzymes at the heart of metabolism.

## Steady state existence and stability analysis of a simple autocatalytic cycle

To explore general principles governing the dynamic behavior of autocatalytic cycles, we consider the simple autocatalytic cycle depicted in *Figure 3A*. This cycle has a single intermediate metabolite, $X$. We denote the flux through the autocatalytic reaction of the cycle by $f_a$, such that for any unit of $X$ consumed, it produces two units of $X$. The autocatalytic reaction assimilates an external metabolite (denoted $A$), which we assume to be at a constant concentration. We denote the flux through the reaction branching out of the cycle by $f_b$. Biologically, $f_b$ represents the consumption of the intermediate metabolite $X$. In the cycles we find in central carbon metabolism, the branch reactions provide

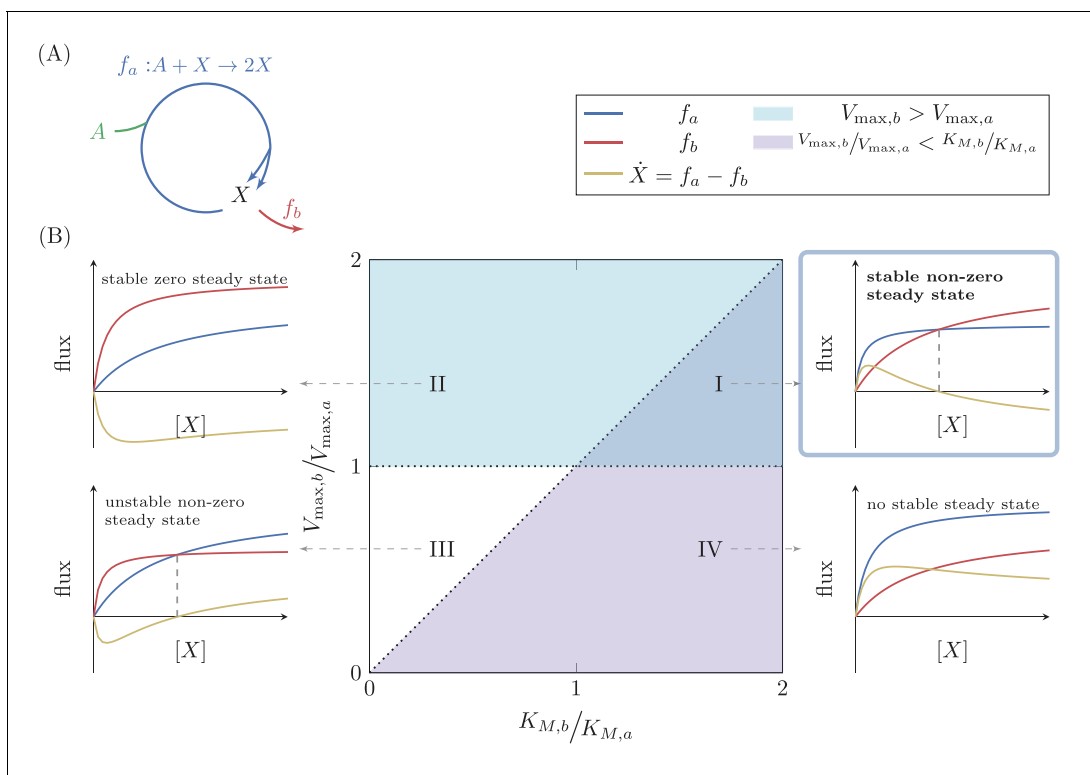

**Figure 3.** Analysis of a simple autocatalytic cycle. (A) A simple autocatalytic cycle induces two fluxes, $f_a$ and $f_b$ as a function of the concentration of $X$. These fluxes follow simple Michaelis-Menten kinetics. A steady state occurs when $f_a = f_b$, implying that $\dot{X} = 0$. The cycle always has a steady state (i.e. $\dot{X} = 0$) at $X = 0$. The slope of each reaction at $X = 0$ is $V_{max}/K_M$. A steady state is stable if at the steady state concentration $\frac{d\dot{X}}{dX} < 0$. (B) Each set of kinetic parameters, $V_{max,a}, V_{max,b}, K_{M,a}, K_{M,b}$ determines two dynamical properties of the system: If $V_{max,b} > V_{max,a}$, then a stable steady state concentration must exist, as for high concentrations of $X$ the branching reaction will reduce the concentration of $X$ (cyan domain, cases (I) and (II)). If $\frac{V_{max,b}}{K_{M,b}} < \frac{V_{max,a}}{K_{M,a}}$, implying that $\frac{V_{max,b}}{V_{max,a}} < \frac{K_{M,b}}{K_{M,a}}$, then zero is a non-stable steady state concentration as if $X$ is slightly higher than zero, the autocatalytic reaction will carry higher flux, further increasing the concentration of $X$ (magenta domain, cases (I) and (IV)). At the intersection of these two domains a non-zero, stable steady state concentration exists, case (I).

precursors that support growth through subsequent reactions. We thus also sometimes consider $f_b$ to represent biomass generation.

For simplicity in the derivation, we assume irreversible Michaelis-Menten kinetics for the two reactions. Even though $f_a$ should follow bisubstrate velocity equation, assuming constant concentration of $A$ reduces the bisubstrate equation to a simple Michaelis-Menten equation. The apparent kinetic constants of the equation depend on the constant value of $A$ (see Materials and methods section "Connecting bisubstrate reaction kinetic constants with simple Michaelis-Menten constants"). We extend our analysis to bisubstrate reaction equations in the next section. We therefore assume that:

$$f_a = \frac{V_{\max,a} X}{K_{M,a} + X}$$

$$f_b = \frac{V_{\max,b} X}{K_{M,b} + X}$$

where $V_{\max}$ is the maximal flux each reaction can carry and $K_M$ is the substrate concentration at which half the maximal flux is attained. Physiologically, these kinetic parameters must be positive. Using these simple forms allows us to obtain an analytic solution. We discuss more general cases below.

We characterize the metabolic state of this system by the concentration of the metabolite $X$. We note that knowing the concentration of $X$ suffices in order to calculate the fluxes originating from it, $f_a$ and $f_b$, thus fully defining the state of the system. A steady state of the system is defined as a concentration, $X^*$, which induces fluxes that keep the concentration constant, such that the total in-flux to $X$ is exactly equal to the total out-flux from it. In our example, the outgoing flux from $X$ is $f_a + f_b$ and the incoming flux to $X$ is $2f_a$, so at steady state it holds that:

$$\dot{X} = \frac{dX}{dt} = 2f_a - (f_a + f_b) = 0 \tag{1}$$

Intuitively, at steady state, the branch reaction must consume all the excess intermediate metabolite that is produced by the autocatalytic reaction. Indeed, expanding the condition above gives:

$$f_a = f_b \Rightarrow \frac{V_{\max,a} X^*}{K_{M,a} + X^*} = \frac{V_{\max,b} X^*}{K_{M,b} + X^*}$$

which is satisfied either if $X^* = 0$ or if:

$$X^* = \frac{V_{\max,b} K_{M,a} - V_{\max,a} K_{M,b}}{V_{\max,a} - V_{\max,b}} \tag{2}$$

implying that:

$$\frac{X^*}{K_{M,a}} = \frac{\frac{V_{\max,b}}{V_{\max,a}} - \frac{K_{M,b}}{K_{M,a}}}{1 - \frac{V_{\max,b}}{V_{\max,a}}} \tag{3}$$

The concentration of $X$ cannot be negative, and thus we get a constraint on the kinetic parameters for which a positive steady state exists. Either both the numerator and the denominator of **Equation 3** are positive, such that:

$$1 > \frac{V_{\max,b}}{V_{\max,a}} > \frac{K_{M,b}}{K_{M,a}},$$

or both are negative, such that:

$$1 < \frac{V_{\max,b}}{V_{\max,a}} < \frac{K_{M,b}}{K_{M,a}}$$

These constraints are graphically illustrated in **Figure 3B**, cases (III) and (I).

In order to gain intuition for this relationship we note that $\frac{V_{\max}}{K_m}$ is the slope of the Michaelis Menten function at $X = 0$. The existence of a positive steady state can be used to get that:

$$X^* > 0 \Rightarrow \frac{V_{\max,b}K_{M,a} - V_{\max,a}K_{M,b}}{V_{\max,a} - V_{\max,b}} > 0 \Rightarrow \frac{\frac{V_{\max,b}}{K_{M,b}} - \frac{V_{\max,a}}{K_{M,a}}}{V_{\max,a} - V_{\max,b}} > 0$$

The last inequality above implies that in order for a positive steady state to exist, the reaction with higher maximal flux must have a shallower slope at $X = 0$. Mathematically, the constraint states that if $V_{\max,a} > V_{\max,b}$ then $\frac{V_{\max,a}}{K_{M,a}} < \frac{V_{\max,b}}{K_{M,b}}$. Alternatively, if $V_{\max,a} < V_{\max,b}$ then $\frac{V_{\max,a}}{K_{M,a}} > \frac{V_{\max,b}}{K_{M,b}}$. This condition can be intuitively understood, as the reaction with shallower slope at $X = 0$ has smaller fluxes for small values of $X$ compared with the other reaction, so unless it has higher fluxes than the other reaction for large values of $X$ (meaning that its maximal flux is higher), the two will never intersect (see *Figure 3B*).

While having a steady state at positive concentration is an essential condition to sustain flux, it is not sufficient in terms of biological function. The steady state at positive concentration must also be stable to small perturbations. Stability with respect to small perturbations is determined by the response of the system to small deviations from the steady state, $X^*$ (at which, by definition $\dot{X} = 0$). Assuming $X = X^* + \Delta X$, stability implies that if $\Delta X$ is positive then $\dot{X}$ needs to be negative at $X^* + \Delta X$, reducing $X$ back to $X^*$, and if $\Delta X$ is negative, $\dot{X}$ will need to be positive, increasing $X$ back to $X^*$. It then follows that in order for $X^*$ to be stable, $\frac{d\dot{X}}{dX} < 0$ at $X = X^*$, implying that upon a small deviation from the steady state $X^*$ (where $\dot{X} = 0$), the net flux $\dot{X}$ will oppose the direction of the deviation.

For the simple kinetics we chose, the stability condition dictates that:

$$\frac{d\dot{X}}{dX}\bigg|_{X=X^*} = \frac{V_{\max,a}K_{M,a}}{(K_{M,a} + X^*)^2} - \frac{V_{\max,b}K_{M,b}}{(K_{M,b} + X^*)^2} < 0 \tag{4}$$

The analysis is straightforward for the case of $X^* = 0$, yielding that 0 is a stable steady state concentration if $\frac{V_{\max,b}}{K_{M,b}} > \frac{V_{\max,a}}{K_{M,a}}$, corresponding to the area above the diagonal in *Figure 3B*, where $\frac{V_{\max,b}}{V_{\max,a}} > \frac{K_{M,b}}{K_{M,a}}$. These cases are denoted as cases (II) and (III). If the relation is reversed (i.e. $\frac{V_{\max,b}}{K_{M,b}} < \frac{V_{\max,a}}{K_{M,a}}$), then 0 is an unstable steady state. The criterion that is of interest, however, is the criterion for stability of the non-zero steady state, $X^* = \frac{V_{\max,b}K_{M,a} - V_{\max,a}K_{M,b}}{V_{\max,a} - V_{\max,b}}$. In this case, substituting $X^*$ in *Equation 4* gives the opposite condition to that of $X^* = 0$. This steady state is thus stable if $\frac{V_{\max,b}}{K_{M,b}} < \frac{V_{\max,a}}{K_{M,a}}$, corresponding to the magenta domain in *Figure 3B*, containing cases (I) and (IV), and unstable otherwise.

The stability criterion can be generally stated in metabolic control terms (*Fell, 1997*) using the notion of elasticity coefficients of reactions, defined as:

$$\varepsilon_X^f = \frac{\partial f}{\partial X}\frac{X}{f}$$

In these terms, stability is obtained if and only if the elasticity of the branch reaction at the positive steady state concentration is greater than the elasticity of the autocatalytic reaction:

$$\frac{df_b}{dX}\bigg|_{X=X^*} > \frac{df_a}{dX}\bigg|_{X=X^*} \Rightarrow \varepsilon_X^{f_b} > \varepsilon_X^{f_a}$$

The complete analysis is summarized in *Figure 3B*. Domain (I) is the only domain where a positive, stable steady state exists. Domains (I) and (III) are the domains at which a positive steady state concentration exists, but in domain (III) that steady state is not stable. The domains below the diagonal (cases (I) and (IV)) are the domains where $X^* = 0$ is an unstable steady state concentration, so that if another steady state exists, it is stable, but in domain (IV) no positive steady state exists. The domains above the diagonal (cases (II) and (III)) are the domains where $X^* = 0$ is a stable steady state concentration, so that the other steady state, if it exists, is unstable.

Aside from existence and stability, a quantitative relationship between the affinity of the biomass generating, branching reaction and the flux it carries can be obtained. This relationship is opposite to the standard one, meaning that unlike the common case where the flux $f$ increases when the affinity becomes stronger, in this case, because the steady state concentration increases when $K_{M,b}$ becomes weaker (*Equation 7* in Materials and methods section "Steady state concentration

dependence on kinetic parameters of autocatalytic and branch reactions"), $f_b$ also increases when $K_{M,b}$ becomes weaker.

To conclude, for this simple cycle, we get that in order for a positive-concentration stable steady state to exist (case (I)), two conditions must be satisfied:

$$\begin{cases} V_{\max,b} > V_{\max,a} \\ \frac{V_{\max,b}}{K_{M,b}} < \frac{V_{\max,a}}{K_{M,a}} \end{cases} \qquad (5)$$

The first requirement states that the maximal flux of the biomass generating, branching reaction should be higher than the maximal flux of the autocatalytic reaction. This requirement ensures a stable solution exists, as large concentrations of $X$ will result in its reduction by the branch reaction. The second requirement states that for concentrations of $X$ that are close enough to 0, the autocatalytic reaction is higher than the branch reaction (as can be inferred from the slopes). This requirement implies that the two fluxes will be equal for some positive concentration of $X$, ensuring a positive steady state exists. As this requirement further implies that below the positive steady state the branch reaction will carry less flux than the autocatalytic reaction, it follows that small deviations of the concentration of $X$ below the steady state will result in an increase in its concentration by the autocatalytic reaction, driving it back to the steady state. Meeting the second constraint has another consequence.

Interestingly, these conditions imply that if $K_{M,b} < K_{M,a}$ then no positive stable steady state can be achieved. Specifically, changes to the expression levels of the enzymes catalyzing $f_a$ and $f_b$ only affect $V_{\max,a}$ and $V_{\max,b}$, and therefore do not suffice to attain a stable positive steady state. This indicates that stability of autocatalytic cycles, that are represented by the model analyzed above, depends on inherent kinetic properties of the enzymes involved and cannot always be achieved by modulating expression levels. We suggest this property to be a design principle that can be critical in metabolic engineering.

## Integrating the bisubstrate nature of the autocatalytic reaction into the simple model

In the model above, to keep the analysis concise, we neglected the bisubstrate nature of the autocatalytic reaction. We extend the analysis to the most common classes of bisubstrate reaction mechanisms in the Materials and methods, sections "Connecting bisubstrate reaction kinetic constants with simple Michaelis-Menten constants", "Constraints on concentration of assimilated metabolite and kinetic constants of bisubstrate reactions", and "Dependence of steady state concentration on assimilated metabolite". All bisubstrate reaction schemes analyzed take a Michaelis-Menten like form once the concentration of the assimilated metabolite is kept constant (*Equations 8, 10, 12, and 14* in Materials and methods section "Connecting bisubstrate reaction kinetic constants with simple Michaelis-Menten constants").

For any set of kinetic parameters, under all ternary enzyme complex schemes, a lower bound on the concentration of $A$ exists, under which the conditions for the existence and stability of a positive steady state cannot be satisfied (*Equations 18 and 23* in Materials and methods section "Constraints on concentration of assimilated metabolite and kinetic constants of bisubstrate reactions"). The exact value of the minimal concentration of $A$ depends on the specific bisubstrate reaction scheme and the kinetic parameters of it.

In the simplified model analyzed above, stability implied the affinity of the branch reaction towards its substrate was limited. A similar limit exists in most cases of bisubstrate reaction schemes (*Equations 16, 19, and 21* in Materials and methods section "Constraints on concentration of assimilated metabolite and kinetic constants of bisubstrate reactions"). Interestingly, if the bisubstrate reaction is ordered with the internal metabolite binding first, then no strict constraints exist on $K_{M,b}$ and a stable steady state solution can always be achieved by setting appropriate values to $V_{\max,b}$ and $V_{\max}$, the maximal flux of the bisubstrate autocatalytic reaction.

Finally, regarding the dynamic behavior of the system when the concentration of $A$ varies, we note that in all three ternary enzyme complex cases, as the concentration of $A$ approaches its lower bound, the steady state concentration of $X$ approaches 0, reducing both the autocatalytic and the branch fluxes (*Equations 24, 25, and 26* in Materials and methods section "Dependence of steady

state concentration on assimilated metabolite"). In the substituted enzyme mechanism, the lower bound on the concentration of $A$ is 0, at which the steady state concentration of $X$ is trivially 0 as well. In all cases, if the maximal flux of the autocatalytic reaction is higher than the maximal flux of the branch reaction, an upper bound on the concentration of $A$ may also exist, to satisfy the condition that $V_{\max,a} < V_{\max,b}$. However, this bound can be removed by increasing $V_{\max,b}$ or reducing $V_{\max}$.

## Extensions of the simple autocatalytic cycle model

### Generalizing for different autocatalytic stoichiometries

Our didactic analysis considered an autocatalytic reaction with 1:2 stoichiometry, such that for every substrate molecule consumed, two are produced. Real-world autocatalytic cycles may have different stoichiometries. For example, the CBB cycle has a stoichiometry of 5:6 so that for every 5 molecules of five-carbon sugar that the autocatalytic reaction consumes, 6 five-carbon molecules are produced. We can generalize our analysis by defining a positive $\delta$ such that for every molecule of $X$ that $f_a$ consumes, it produces $1 + \delta$ molecules of $X$, where $\delta$ may be a fraction. This extension implies that *Equation (1)* becomes:

$$\dot{X} = \frac{dX}{dt} = (1+\delta)f_a - (f_a + f_b) = 0 \Rightarrow \delta \cdot f_a = f_b \Rightarrow \frac{\delta \cdot V_{\max,a} X}{K_{M,a} + X} = \frac{V_{\max,b} X}{K_{M,b} + X}$$

Therefore, all of the results above can be extended to different stoichiometries by replacing $V_{\max,a}$ with $\delta \cdot V_{\max,a}$. As a result, the qualitative conditions and observations from the 1:2 stoichiometry case remain valid but with a constant factor that changes the quantitative relations according to the relevant stoichiometry.

### Input flux increases the range of parameters for which a stable steady state solution exists

Autocatalytic cycles are embedded within a larger metabolic network. Specifically, such cycles may have independent input fluxes to some of their intermediate metabolites, not requiring the use of other intermediate metabolites of the cycle. For example, in the glucose based, PTS-dependent autocatalytic cycle, the existence of alternative transporters can generate flux of glucose 6-phosphate into the cycle without the use of pep (*Ferenci, 1996*).

When adding a constant input flux, $f_i$ to our simple system (*Figure 4A*) the steady state condition changes to include this flux, giving:

$$\dot{X} = \frac{dX}{dt} = f_i + f_a - f_b = 0$$

In this situation, at $X = 0$, $\dot{X} = f_i > 0$ so the concentration of $X$ increases and there is no steady state at zero. If $V_{\max,b} > f_i + V_{\max,a}$ then at a large enough value of $X$, $\dot{X}$ will be negative, implying that at some value of $X$ between these two extremes, $\dot{X}$ attains the value of zero, such that under this condition a positive stable steady state concentration exists (*Figure 4I*). This case therefore differs from the case with no input flux analyzed above, as now a positive stable steady state can always be achieved by modifying only $V_{\max,a}$ and/or $V_{\max,b}$. In this setup, cells can therefore tune the expression levels of enzymes to meet the needs of a stable steady state flux.

In cases where $V_{\max,b} < f_i + V_{\max,a}$ either no steady states exist (*Figure 4II*), or two positive steady states exist (*Figure 4III*). The latter case implies that there exists a positive concentration $X$ that satisfies:

$$\dot{X} = 0 \Rightarrow f_i + f_a(X) - f_b(X) = 0 \Rightarrow f_i + \frac{V_{\max,a} X}{K_{M,a} + X} = \frac{V_{\max,b} X}{K_{M,b} + X}$$

In this case, the lower concentration steady state will be stable.

To conclude, input fluxes change the steady state(s) of autocatalytic cycles. When an input flux is present, an autocatalytic cycle can always achieve a non zero, stable steady state by tuning the expression levels of the enzymes forming the cycle.

Interestingly, we find that in the two autocatalytic cycles shown in *Figure 2—figure supplements 1* and *2*, reactions that generate direct input flux into the cycle exist. In the ribose-5P assimilating

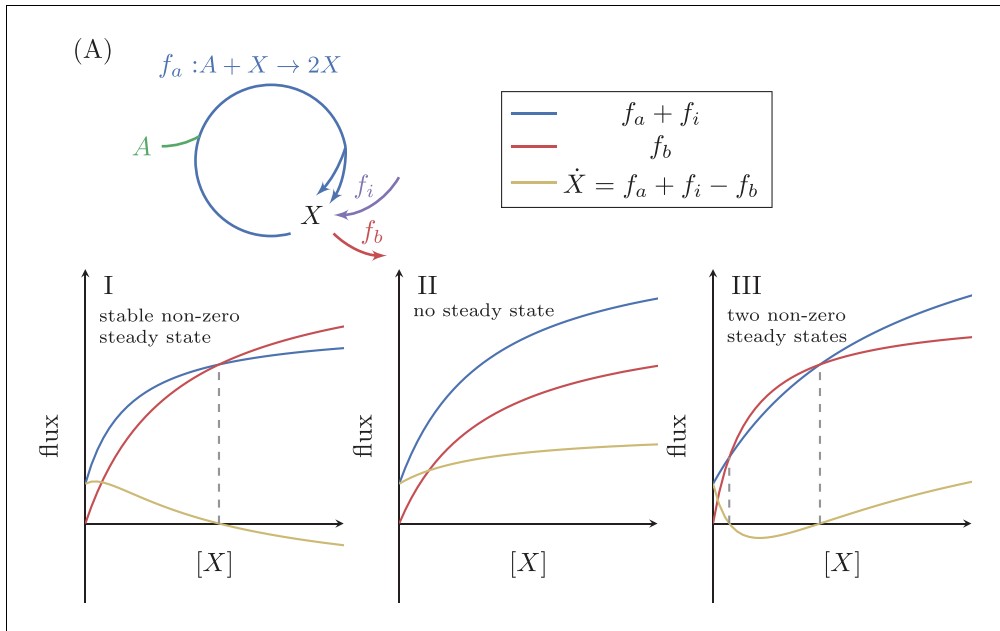

**Figure 4.** Analysis of an autocatalytic cycle with input flux. (A) The effect of a fixed input flux, $f_i$, on the possible steady states of a simple autocatalytic cycle. A steady state occurs when $f_a + f_i = f_b$. If $V_{max,b} > V_{max,a} + f_i$ then there is always a single stable steady state (I). If $V_{max,b} < V_{max,a} + f_i$ then there can either be no steady states (II), or two steady states where the smaller one is stable (III).

autocatalytic cycle (*Figure 2—figure supplement 1*), the rpi reaction serves as a shortcut, allowing input flux directly from ribose-5P into the cycle. In the glycerone-phosphate assimilating cycle (*Figure 2—figure supplement 2*), the tpi reaction similarly serves as such a shortcut. Interestingly, in these two cases, these shortcuts relax the constraints imposed by the strict use of the corresponding autocatalytic cycles as they prevent zero from being a stable steady state concentration. Another example of the effects of the addition of an input flux to an autocatalytic cycle is the input flux of fructose-6-phosphate from the catabolism of starch into the CBB cycle. This input flux can be used to 'kick start' the cycle even without using the intermediate metabolites of the cycle.

## Reversible branch reaction can either be far from equilibrium, resulting in the simple case, or near equilibrium, pushing the stability conditions down the branch pathway

The simple model assumed both the autocatalytic and the branch reactions are irreversible under physiological conditions. Assuming the branch reaction, $f_b$, can be reversible, with a product $Y$, the system can be analyzed in two extreme cases.

If $Y$ is consumed very rapidly by subsequent reactions, keeping its concentration low, then $f_b$ operates far from equilibrium. In this case, the reversible reaction equation reduces to an irreversible Michaelis-Menten equation, resulting in the same constraints as in the simple, irreversible case analyzed above.

If $Y$ is consumed very slowly, and if the maximal consumption of $Y$ is larger than $V_{max,a}$, then, as long as $V_{max,b} > V_{max,a}$, a stable steady state exists both when $V_{max,b} \to \infty$, making $f_b$ operate near equilibrium, and when $V_{max,b} \to V_{max,a}$. A mathematical analysis is provided in the Materials and methods section "Reversible branch reaction analysis". The assumptions on the consumption of $Y$ in this case are analogous to the constraints in *Equation 5*, namely that the reaction downstream of $Y$ is less saturated than the autocatalytic reaction, and that it consumes $Y$ at a lower rate than the rate at which the autocatalytic reaction produces $X$ near $X = 0$.

## Analysis of a reversible autocatalytic cycle reaction

The simple model assumed both the autocatalytic and the branch reactions are irreversible under physiological conditions. The autocatalytic reaction in the simple model represents an effective over-all reaction for all of the steps in autocatalytic cycles found in real metabolic networks. In order for the combined autocatalytic reaction to be physiologically reversible, all of the reactions in the real metabolic network must be reversible under physiological conditions. We note that this is not the case in any of the cycles we identify in central carbon metabolism. Nevertheless, this case can be mathematically analyzed.

If the autocatalytic reaction is reversible, then it must be driven by the displacement from thermodynamic equilibrium of the concentration of $A$ versus the concentration of $X$. Therefore, for any fixed concentration of $A$, $\hat{A}$, a concentration of $X$ exists such that $f_a(X, \hat{A}) = 0 < f_b(X)$. It then follows that a sufficient condition for a positive steady state to exist is that at $X = 0$, $\dot{X}(\hat{A}) > 0$, which implies that

$$\left. \frac{\partial f_a}{\partial X} \right|_{X=0, A=\hat{A}} > \frac{V_{\max,b}}{K_{M,b}}$$

This condition can always be satisfied by high expression of the autocatalytic enzyme, increasing $V_{\max,a}$. For this case, it therefore follows that for any concentration of $A$, a minimal value for $V_{\max,a}$ exists, above which a positive steady state is achieved.

## Stability analysis for multiple-reaction cycles

Even the most compact real-world autocatalytic cycles are composed of several reactions. It is thus useful to extend the simple criteria we derived to more complex autocatalytic cycles. In such cycles the criteria for the existence of a steady state become nuanced and detail specific. We therefore focus on evaluating stability of such cycles, under the assumption that a non-zero steady state exists, which is usually known based on experimental measurements.

We analyze the stability criteria for the autocatalytic cycles depicted in **Figure 5A and B** in the Materials and methods, section "Extending the stability analysis from single to multiple reaction cycles". The analysis is performed for autocatalytic ratios up to 1:2, which is the case for all the

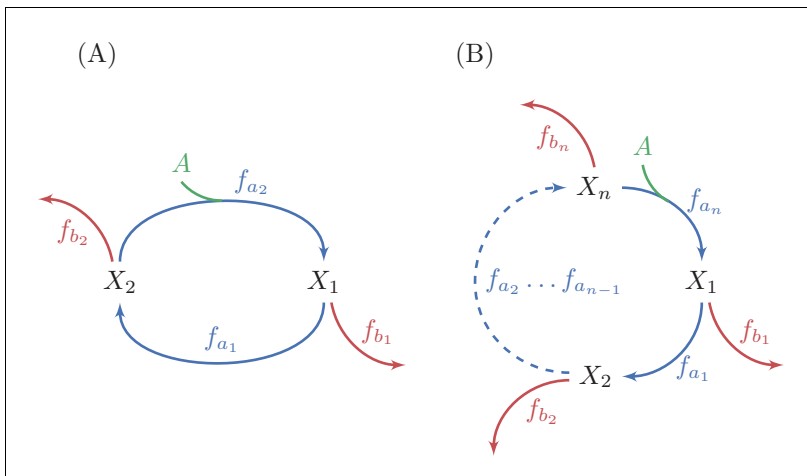

**Figure 5.** Generalization of analysis to multiple-reaction autocatalytic cycles with a single assimilating reaction. (**A**) A two reaction system. (**B**) A generic $n$-reaction system. The system is at steady state when the total consumption of intermediate metabolites by the branch reactions is equal to the flux through the autocatalytic reaction, because the autocatalysis is in a 1:2 ratio. A sufficient condition for the stability of a steady state in these systems is that the derivative of at least one branch reaction with respect to the substrate concentration is larger than the derivative of the equivalent autocatalytic reaction at the steady state concentration. Given the connection between derivatives of fluxes and saturation levels of reactions (see methods), this condition implies that at a stable steady state, the saturation level of at least one branch reactions is smaller than the saturation level of the corresponding autocatalytic reaction.

autocatalytic cycles we identify. We find that in the multiple reaction case a steady state is stable if there exists $i$ such that $\beta_i > \alpha_i$ (where $i$ can be any number in the range $1 \ldots n$, $\alpha_i = \frac{\partial f_{a_i}}{\partial X_i}$, and $\beta_i = \frac{\partial f_{b_i}}{\partial X_i}$), Materials and methods section "Limits on derivatives of branch reactions for complex autocatalytic cycles".

Using the connection between derivatives of reactions and saturation levels (Materials and methods section "Inverse relationship between derivatives, affinities, and saturation levels"), we conclude that if $\beta_i > \alpha_i$ for some $i$ at the steady state point, then the saturation of the branch reaction, denoted $S(f)$, must be lower than the saturation of the corresponding cycle reaction at $X_i$:

$$S(f_{b_i}) < S(f_{a_i}) \tag{6}$$

This condition also dictates that the affinity of the branch reaction to the intermediate metabolite of the cycle it consumes must be weaker than the affinity of the corresponding recycling reaction of the cycle.

While having a single branch point at which $\beta_i > \alpha_i$ is a sufficient condition for stability, we note that the larger the number of branch points satisfying this condition, the more robust the steady state point will be to perturbations, as such branch points reduce the propagation of deviations along the cycle (see Materials and methods section "Multiple unsaturated branch reactions increase convergence speed and dampen oscillations"). As we show below, these predictions hold for functioning autocatalytic cycles.

## Using different kinetic equations

Although we utilized the widely-used irreversible Michaelis-Menten kinetics equation to model enzyme kinetics, our results can be extended to different kinetic equations. Generally, two conditions must be met for a stable flux through an autocatalytic cycle to exist: (A) there should be a positive concentration of the intermediate metabolites for which the outgoing fluxes balance the autocatalytic fluxes, resulting in a steady state, and, (B) at the steady state point at least one derivative of an outgoing reaction out of the cycle should be higher than the derivative of the corresponding cycle reaction, as is implied by *Equation 31*, to enforce stability in the presence of small perturbations. Therefore, these two conditions should be explicitly evaluated for every case with different kinetic equations and autocatalytic cycles topologies to assert whether it can carry stable fluxes or not.

## Testing the predictions of the analysis with experimental data on functioning autocatalytic cycles

To evaluate the validity of our analysis of autocatalytic cycles we searched for growth conditions under which the autocatalytic cycles we identified in central carbon metabolism carry substantial flux in-vivo. We used recent in-vivo flux measurements in *E. coli* from *Gerosa et al. (2015)*. According to the data, two autocatalytic cycles carry substantial flux under at least one of the growth conditions measured: a cycle using the PTS carries significant fluxes in growth on glucose and on fructose; the glyoxylate cycle carries significant flux in growth on acetate and on galactose.

As noted above, we predict a design principle for functioning autocatalytic cycles: that at least one branch reaction should have a steeper response than the corresponding autocatalytic reaction at steady state. This requirement is sufficient, but not necessary, for the autocatalytic cycle to be at a stable steady state point. Moreover, having more than one branch point at which the branch reaction has a steeper response than the autocatalytic reaction increases the robustness of the steady state flux in the cycle as shown in the Materials and methods section "Multiple unsaturated branch reactions increase convergence speed and dampen oscillations". An outcome of the relationship between the steepness's of responses is a reverse relationship between the saturation levels of the corresponding reactions (*Equation 35*). In order to evaluate the saturation level of a reaction under a given condition, two values must be obtained:

1. The maximal flux capacity of the reaction under the given condition, $V_{\max}$.
2. The actual flux through the reaction at the steady state, $f$.

To estimate the maximal capacity of a reaction we followed the procedure described in (*Davidi et al., 2016-22*) (see Materials and methods section "Evaluating maximal flux capacity of reactions under a given condition"). We used the data from (*Gerosa et al., 2015*) to identify the major branch points in each functioning cycle and the actual flux in them under each of the relevant conditions. The results are presented in *Figure 6* and are provided, with the relevant calculations, in *Supplementary file 1*.

Our results show that for any of the four functioning autocatalytic cycle cases, in at least one branch point the biomass generating branch reaction has a larger maximal flux capacity, and is considerably less saturated than the respective autocatalytic reaction, in accordance with our predictions. Moreover, out of nine branch points analyzed, in six branch points the branching reactions were significantly less saturated than the autocatalytic reactions, in two branch points the saturation levels were similar, and only in one branch point the autocatalytic reaction was less saturated than the branching reaction.

The branch point at which the autocatalytic reaction is less saturated than the branch reaction is the branch point from fructose-1,6-bisphosphate in growth on fructose as the carbon source. The high saturation of the branch reaction arises as a large flux is reported for the fbp reaction, whereas the corresponding enzyme is not highly expressed under this condition. The large reported flux through fbp arises due to assuming a single transport pathway for fructose, entering the cycle as

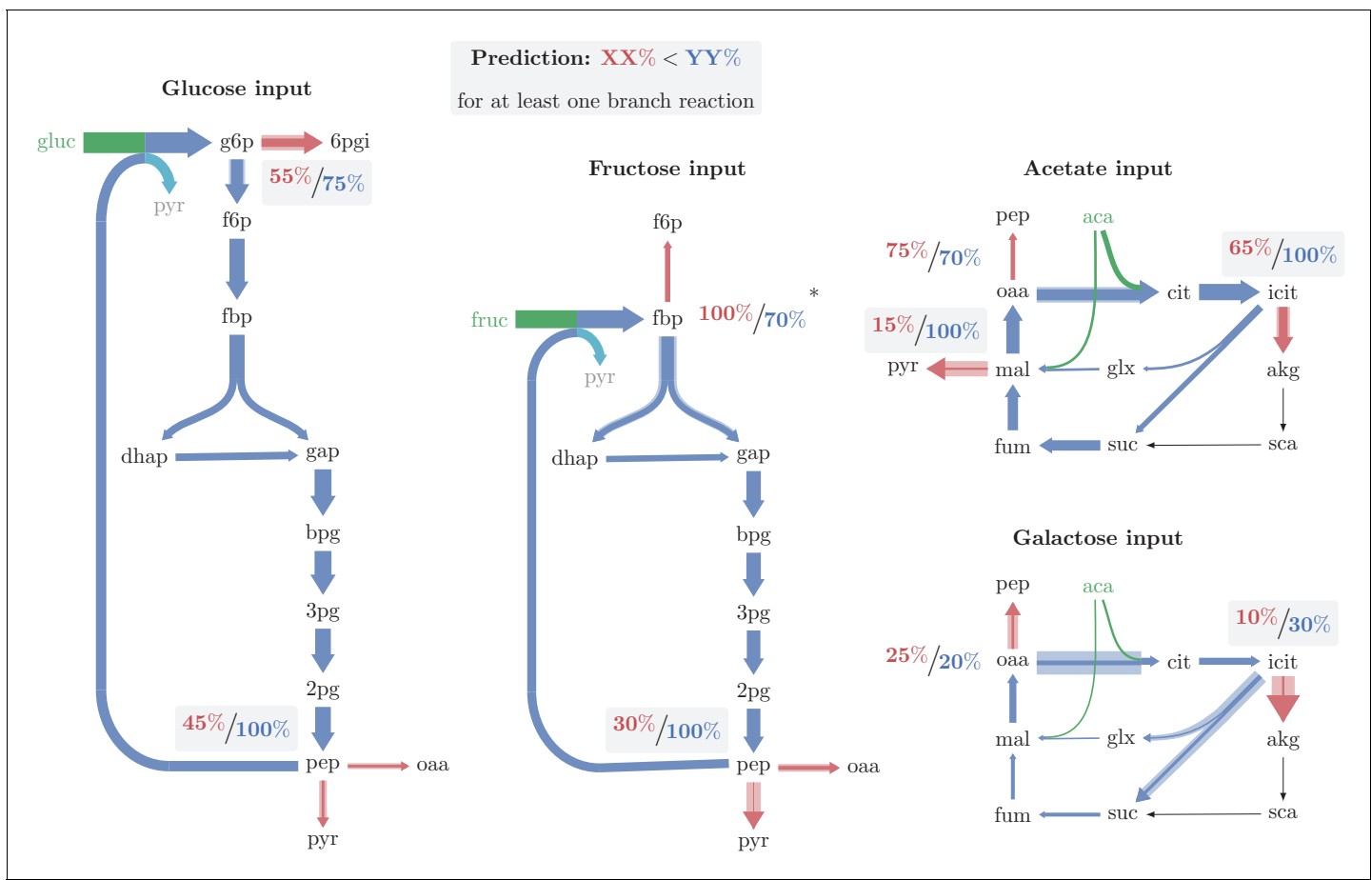

**Figure 6.** Major branch points and relative enzyme saturation in operating autocatalytic cycles. Solid arrow width represents carbon flux per unit time. Shaded arrow width represents the maximal carbon flux capacity per unit time, given the expression level of the catalyzing enzyme. In all cases there is enough excess capacity in the branching reactions to prevent the cycle from overflowing. A 4% flux from pep to biomass was neglected in growth under glucose and fructose. Only in one out of the nine branch points observed (the branch point at fbp in growth under fructose), the outgoing reaction is significantly more saturated than the autocatalytic reaction. (*) A branch point at which the branching reaction is more saturated than the autocatalytic reaction, which may result from neglecting fructose transport directly as f6p when deriving fluxes (see text).

fructose-1,6-bisphosphate. However, an alternative fructose transport pathway is known to occur for the concentration at which the measurements were made (*Kornberg, 1990*). The alternative transport pathway produces fructose-6-phosphate from external fructose. Therefore, any flux through the alternative transport pathway should be directly deducted from the flux through fbp. Assuming 20% of the consumed fructose uses this pathway suffices in order to balance the saturation levels at the fructose-1,6-bisphosphate branch point.

We made two negative control analyses to examine whether other reasons do not underlie the trend we find. First, we compared the saturation levels at the same branch points in growth conditions at which the autocatalytic cycles do not function, but the reactions carry flux. We find that for these cases, only 4 out of 9 cases satisfy the low branch saturation condition (*Supplementary file 1*). Second, we searched for branch points out of non-autocatalytic cycles and tested whether in such points branch reactions are also consistently less saturated than their corresponding cycle reactions. We found two flux-carrying cycles: the TCA cycle, carrying flux in glucose, fructose, and glycerol growth, and a cycle consisting of the pentose-phosphate pathway combined with gluconeogenesis, carrying flux in acetate, glycerol, and succinate growth. Out of the total six conditions-branch points cases, in three the branch reaction was less saturated than the cycle reaction, and in three the cycle reaction was less saturated than the branch reaction (*Supplementary file 1*). We therefore conclude that, for cases that do not involve autocatalysis, the saturation of branch versus cycle reactions seems evenly distributed.

The consistently lower saturation values of biomass generating branch reactions demonstrate that the expressed enzymes have enough capacity to prevent the autocatalytic cycle from increasing the concentration of intermediate metabolites infinitely. Moreover, the lower saturation values of the biomass generating reactions suggest that at the steady state point their derivatives are higher, ensuring stable operation of the cycle.

Another demonstration of the autocatalytic mechanism being at play is in the CBB cycle, which is not a part of the metabolic network of wild type *E. coli*, and for which no flux measurements are available. This cycle has been recently introduced synthetically into *E. coli* and was shown to carry flux in it, given further metabolic engineering of central carbon metabolism (*Antonovsky et al., 2016*). The experimentally observed key evolutionary event enabling the functioning of the CBB cycle, was a mutation affecting the kinetic properties of the main branching reaction out of the CBB pathway, prs, weakening its affinity to its substrate, ribose-5p. The observed weakening of affinity of prs is directly in line with our predictions on the relationship between the affinity of branch reactions and the affinity of the corresponding cycle reactions (see Materials and methods section of *Antonovsky et al., 2016*).

The other examples of autocatalytic cycles we found did not carry flux in any of the conditions for which data were available. The pentose-phosphate cycle variants do not carry flux in any of the measured conditions, which is expected given that growth on ribose was not measured. The gluconeogenic FBA with ED pathway cycle also did not carry flux in any of the measured conditions. Although glycerol could have been a potential carbon source to use this pathway, the metabolic network allows for a more energy efficient growth by using the tpi reaction, as was indeed observed.

To conclude, existing data supports predictions made by our model, given the requirement for stable steady state operation of autocatalytic cycles. This agreement between predictions and measurements is especially encouraging given the highly limited information on kinetic properties, concentrations, and fluxes under various growth conditions.

## Analysis of allosteric regulation potential for cycle improvement

Allosteric regulation can modulate the kinetic properties of enzymes at branch points, and of the cycle in general. As such, the relevant condition for the existence of a stable positive steady state should hold for the updated kinetic properties as defined following the effect of allosteric regulation.

We further analyze the ability of specific allosteric interactions to support fast convergence and stability of autocatalytic cycles in the Materials and methods section "Allosteric regulation can improve network performance". We compare the expected beneficial allosteric interactions against the allosteric regulation network of the two functioning autocatalytic cycles we identified, the PTS-using autocatalytic cycle and the glyoxylate cycle (*Supplementary file 1*, regulation data were taken from *Keseler et al. (2013)* and *Schomburg et al. (2004)*).

For the PTS cycle, we find that there are a total of 12 allosteric interactions, 7 inhibitions and five activations. Out of these 12 interactions, 11 interactions follow our expectations in terms of the type of the regulating metabolite (assimilated metabolite, cycle intermediate, or branch product), the regulated reaction (cycle reaction, branch reaction, or the reverse of a branch reaction), and the direction of the regulation (activation or inhibition). One interaction, the activation of fba by pep, does not follow our expectation.

For the glyoxylate cycle, we find that there are a total of 13 interactions, 12 inhibitions and one activation. Out of these 13 interactions, 8 interactions follow our expectations and 5 do not. The lack of significant agreement between the expected regulation direction and the actual regulation found for this cycle is consistent with the observation in *Gerosa et al. (2015)* that TCA cycle fluxes are regulated mainly by transcription and not by reactants levels.

It is important to note that allosteric regulation serves many roles, and that the metabolic network faces many more challenges than just the support of stable autocatalysis. Therefore, the agreement we find between existing allosteric interactions and the expected regulation scheme supporting autocatalysis does not suggest that the autocatalytic nature of the PTS is the only, or even main underlying reason for these allosteric interactions.

## Discussion

Our study into the dynamics and stability of autocatalytic cycles suggests design principles applicable to both systems biology, that aims to understand the function of natural networks, and in the context of synthetic biology, in the effort to express novel heterologous cycles.

While autocatalytic cycles are often overlooked in the study of metabolism, we find that such cycles are at the heart of central carbon metabolism. Our autocatalytic modeling framework gives concrete predictions on saturation levels of branch reactions for operating autocatalytic cycles. We find these predictions agree well with empirically measured fluxomics and proteomics data sets. Given that there are other suggestions (*Staples and Suarez, 1997*; *Weiss et al., 1998*; *Suarez et al., 1997*) that may underlie the low saturation of branch reactions, we compare the saturation levels of branch reactions versus their corresponding cycle reactions both under conditions when the autocatalytic cycle does not function, and for branch points out of non-autocatalytic cycles. Both tests show no bias towards low saturation of branch reactions out of non-autocatalytic cycles, contrary to the clear trend we find for reactions branching out of autocatalytic cycles. Our findings thus support the addition of stability of intermediate metabolites of autocatalytic cycles as an explanation for the seemingly wasteful expression of enzymes (*Salvador and Savageau, 2003, 2006*). The model we present can also highlight metabolic branch points at which the kinetic efficiency of enzymes is constrained due to stability requirements of a corresponding autocatalytic cycle.

A common concept in synthetic biology is that the successful implementation of novel pathways requires the expression of functional enzymes in the correct amounts in the target organism. Here we show that in the context of autocatalytic cycles, such expression modulation may not suffice. Specifically, changes to the substrate affinity of enzymes at branch points of the cycle may be required in order for the novel pathway to function.

Another aspect of our findings is that while it is common to assume that strong affinity and high catalytic rate are desirable traits for enzymes, such seeming improvements may actually lead to instability and thus to non functional metabolic cycles. Furthermore, for reactions branching out of autocatalytic cycles, weaker affinities increase the steady state concentration of intermediate metabolites, resulting in higher fluxes both through the cycle, and through the branch reaction, suggesting an unconventional strategy for optimizing fluxes through such reactions. We note that because allosteric regulators modify the affinity of the enzymes they target, such regulators can potentially be used to restrict the affinity of branch reactions only when the autocatalytic cycle functions.

An experimental demonstration of these principles in-vivo is the recent implementation of a functional CBB cycle in *E. coli* by introducing the two genes missing for its function (*Antonovsky et al., 2016*). The successful introduction of the genes did not suffice to make the cycle function, and further directed evolution was needed in order to achieve successful operation of the cycle. Strikingly, most evolutionary changes occurred in branch points from the cycle (*Antonovsky et al., 2016*). The

change which was biochemically characterized in the evolutionary process was the decrease of the value of $\frac{k_{cat}}{K_M}$ of phosphoribosylpyrophosphate synthetase (prs), one of the enzymes responsible for flux out of the CBB cycle, corresponding to the branch reaction in our simple model. This is beautifully in line with the predictions of our analysis that suggest that decreasing $\frac{V_{max}}{K_M}$ of branch reactions can lead to the existence of a stable flux solution.

Our observation regarding the stabilizing effect of input fluxes into an autocatalytic cycle can provide means to mitigate the stability issue in synthetic biology metabolic engineering setups. In such setups, conducting directed evolution under gradually decreasing input fluxes, such as those achieved in a chemostat, allows for a pathway to gradually evolve towards sustainable, substantial flux.

Finally, while our work focuses on cycles increasing the amount of carbon in the system, we note that autocatalysis can be defined with respect to other quantities such as energy (e.g. ATP investment and production in glycolysis [*Teusink et al., 1998*]), non-carbon atoms, reducing power, or other moieties (*Reich and Selkov, 1981*). As autocatalysis is often studied with relation to the origin of life, our analysis may be useful in studying synthetic autocatalytic systems such as the one recently described in *Semenov et al. (2016)*. The analysis we present here can thus be of relevance for the analysis of metabolic networks in existing organisms and for the design of novel synthetic systems.

## Materials and methods

### Formal definition of an autocatalytic metabolic cycle

Given a metabolic network composed of a set of reactions and metabolites, the following criteria can be used to define a subset of the network that is an autocatalytic cycle: First we define a metabolic cycle. A set of irreversible reactions (for reversible reactions only one direction can be included in the set) and metabolites forms a cycle if every metabolite of the set can be converted, by sequential application of reactions in the set (where two reactions can be chained if a metabolite in the set is a product of the first reaction and a substrate of the second reaction), to every other metabolite in the set. A cycle is autocatalytic if the reactions of the cycle can be applied, each reaction at an appropriate, positive number of times, such that the resulting change in the amount of each of the metabolites forming the cycle is non-negative, with at least one metabolite of the cycle having a strictly positive change.

The same definition can be stated in terms of reaction vectors and a stoichiometric matrix. If a metabolic network has $n$ metabolites, indicated by the numbers 1 to $n$, then every reaction, $r$, in the network can be described as a vector $V_r$ in $\mathbb{Z}^n$, such that the $i$'th coordinate of $V_r$ specifies how much of metabolite $i$ the reaction $r$ produces (if $r$ consumes a metabolite, then the value at the coordinate representing the metabolite is negative).

With this notation, a set of metabolites: $M = m_1 \cdots m_j$ and a set of reactions, $R = r_1 \cdots r_k$ form an autocatalytic cycle if:

1. Every row and every column of the stoichiometric matrix have at least one positive and one negative number.
2. There is a set of positive integers, $i_1 \cdots i_k$ such that the total reaction vector $r^* = \sum_{l=1}^{k} i_l r_l$ is non negative at all the coordinates $m_1 \cdots m_j$ and is strictly positive for at least one coordinate in this range.
3. The condition that the total reaction vector $r^* = \sum_{l=1}^{k} i_l r_l$ is non negative at all the coordinates $m_1 \cdots m_j$ and is strictly positive for at least one coordinate in this range cannot be satisfied by a set of non-negative integers, $i_1 \cdots i_k$, if this set includes values that are 0 (this condition eliminates the addition of disjoint cycles to an autocatalytic cycle).

### Systematic identification of autocatalytic cycles in metabolic networks

We implemented an algorithm to systematically search for autocatalytic cycles in metabolic networks. The algorithm is not comprehensive, in the sense that there may be autocatalytic cycles that will not be identified by it. Further work will enable a more advanced algorithm to identify additional autocatalytic cycles in full metabolic networks. We used the algorithm on the core carbon metabolism network of *E. coli* (*Orth et al., 2010*).

In our framework, a metabolic network is defined by a set of reactions, $(\bar{R})$. Each reaction is defined by a set of substrates and a set of products, with corresponding stoichiometries $R_i = (S, P, N^S, N^P)$, such that $R_i$ describes the reaction $\sum_j N_j^S S_j \rightarrow \sum_k N_k^P P_k$. The algorithm works as follows:

1. All co-factors are removed from the description of the metabolic network.
2. The metabolic network is converted to a directed graph, G: The nodes of G are all the metabolites and all the reactions of the network. For each metabolite, $M$, and each reaction, $R$, if $M$ is a substrate of $R$ then the edge $(M, R)$ is added to the graph, and if $M$ is a product of $R$, then the edge $(R, M)$ is added to the graph.
3. The Tarjan cycle identification algorithm is used to enumerate all the cycles in the graph (**Tarjan, 1973**).
4. For every cycle identified by the Tarjan algorithm, $C$, the algorithm checks if the cycle can be the backbone of an autocatalytic cycle as follows:

    a. For every reaction in the cycle, $R$, the algorithm checks if it consumes more than one intermediate metabolite of the cycle. If so, $C$ is assumed not to be autocatalytic and the algorithm continues to evaluate the next cycle.
    b. Otherwise, for every reaction in the cycle, $R$, the algorithm checks if it has more than one product that is an intermediate metabolite of the cycle. If so, then the algorithm lists $C$ as an autocatalytic cycle.
    c. Finally, the algorithm checks, for every reaction in the cycle, if it has a product that is not an intermediate metabolite of the cycle. If so, denote by $M_E$ such a metabolite. The algorithm proceeds to check if, for every intermediate metabolite of the cycle, $M_i$ a reaction exists from $M_E$ to $M_i$ that does not use any of the reactions of the cycle, and does not consume any of the intermediate metabolites of the cycle. If so then the algorithm lists $C$ as an autocatalytic cycle.

The algorithm assumes reactions consume exactly one molecule of any of their substrates and produce exactly one molecule of any of their products, an assumption that holds for the core model of *E.coli*, but not in metabolic networks in general.

## Steady state concentration dependence on kinetic parameters of autocatalytic and branch reactions

The simple cycle steady state concentration, $X^*$, is given in *Equation 2*. Taking the derivative of this expression with respect to $K_{M,a}$, $K_{M,b}$, $V_{\max,a}$, and $V_{\max,b}$, under the assumption that the kinetic parameters satisfy the stability conditions in *Equation 5* gives:

$$
\begin{aligned}
\frac{\partial X^*}{\partial K_{M,a}} &= \frac{V_{\max,b}}{V_{\max,a} - V_{\max,b}} < 0 \\
\frac{\partial X^*}{\partial K_{M,b}} &= \frac{-V_{\max,a}}{V_{\max,a} - V_{\max,b}} > 0 \\
\frac{\partial X^*}{\partial V_{\max,a}} &= \frac{V_{\max,b}(K_{M,b} - K_{M,a})}{(V_{\max,a} - V_{\max,b})^2} > 0 \\
\frac{\partial X^*}{\partial V_{\max,b}} &= \frac{V_{\max,a}(K_{M,a} - K_{M,b})}{(V_{\max,a} - V_{\max,b})^2} < 0
\end{aligned}
\tag{7}
$$

So that $X^*$ increases when $K_{M,a}$ decreases or $V_{\max,a}$ increases (or both) corresponding to activation of $f_a$. On the other hand, $X^*$ decreases when $K_{M,b}$ decreases or $V_{\max,b}$ increases (or both) corresponding to activation of $f_b$.

## Connecting bisubstrate reaction kinetic constants with simple Michaelis-Menten constants

Three standard equations are commonly used to describe the flux through irreversible bisubstrate reactions (**Leskovac, 2003**). We show that, under the assumption that the assimilated metabolite maintains constant concentration, these equations reduce to simple Michaelis-Menten equations. We derive the expressions for the apparent Michaelis-Menten constants, $K_M$ and $V_{\max}$, as functions of the kinetic constants of the bisubstrate reaction and the concentration of the assimilated

metabolite. While the substrates in these equations are generally denoted as $A$ and $B$, here, to keep the notation consistent, we will denote by $A$ the assimilated metabolite and by $X$ the internal metabolite of the cycle.

The simplest equation describing a bisubstrate reaction assumes substituted enzyme (Ping Pong) mechanism (*Imperial and Centelles, 2014*). As this equation is symmetric with respect to the two substrates, we can arbitrarily decide which of the two substrates is the assimilated metabolite, and which is the internal metabolite. We get that the flux through the reaction is:

$$f = \frac{V_{\max}AX}{K_X A + K_A X + AX}$$

Rearranging to get the dependence of the flux on $X$ in a Michaelis-Menten like form we get that:

$$f = \frac{\frac{V_{\max}A}{K_A + A}X}{\frac{K_X A}{K_A + A} + X} \tag{8}$$

which gives apparent Michaelis-Menten kinetic constants of:

$$\begin{aligned} \tilde{V}_{max} &= \frac{V_{\max}A}{K_A + A} \\ \tilde{K}_M &= \frac{K_X A}{K_A + A} \end{aligned} \tag{9}$$

The second bisubstrate reaction mechanism we consider is the ternary enzyme complex with random binding order of the two substrates. As this equation is also symmetric with respect to the two substrates, we can again arbitrarily decide which of the two substrates is the assimilated metabolite, and which is the internal metabolite. We get that the flux through the reaction is:

$$f = \frac{V_{\max}AX}{K_{i,A}K_X + K_X A + K_A X + AX}$$

Rearranging to get the dependence of the flux on $X$ in a Michaelis-Menten like form we get that:

$$f = \frac{\frac{V_{\max}A}{K_A + A}X}{\frac{K_{i,A} + A}{K_A + A}K_X + X} \tag{10}$$

which gives apparent Michaelis-Menten kinetic constants of:

$$\begin{aligned} \tilde{V}_{max} &= \frac{V_{\max}A}{K_A + A} \\ \tilde{K}_M &= \frac{K_{i,A} + A}{K_A + A}K_X \end{aligned} \tag{11}$$

The other equation describing a ternary enzyme complex bisubstrate reaction assumes ordered binding of the substrates. Because in ordered binding the equation is asymmetric with respect to the two substrates, analyzing this reaction is further split according to which of the two substrates is assumed to be the assimilated metabolite with constant concentration.

If the first binding metabolite is assumed to be the assimilated metabolite we get that:

$$f = \frac{V_{\max}AX}{K_{i,A}K_X + K_X A + AX} = \frac{V_{\max}X}{\frac{K_{i,A} + A}{A}K_X + X} \tag{12}$$

which gives apparent Michaelis-Menten kinetic constants of:

$$\begin{aligned} \tilde{V}_{max} &= V_{\max} \\ \tilde{K}_M &= \frac{K_{i,A} + A}{A}K_X \end{aligned} \tag{13}$$

If the first binding metabolite is assumed to be the internal metabolite we get that:

$$f = \frac{V_{\max}AX}{K_{i,X}K_A + K_AX + AX} = \frac{\frac{V_{\max}A}{K_A+A}X}{\frac{K_{i,X}K_A}{K_A+A} + X} \tag{14}$$

which gives apparent Michaelis-Menten kinetic constants of:

$$\begin{aligned} \tilde{V}_{max} &= \frac{V_{\max}A}{K_A + A} \\ \tilde{K}_M &= \frac{K_{i,X}K_A}{K_A + A} \end{aligned} \tag{15}$$

To summarize, the most common equations describing bisubstrate reactions reduce to equations of the same form as Michaelis-Menten equations, under the assumption that one of the metabolites maintains a constant concentration. The apparent kinetic constants of the Michaelis-Menten equivalent equations depend on the kinetic constants of the bisubstrate reactions, as well as on the concentration of the assimilated metabolite.

## Constraints on concentration of assimilated metabolite and kinetic constants of bisubstrate reactions

In *Equation 5* we obtain constraints on the kinetic parameters of Michaelis-Menten reactions that ensure the existence and stability of a positive steady state. We observe that these constraints imply that even if the maximal rates of the two reactions can be easily modified, if $K_{M,b} < K_{M,a}$ then such changes cannot suffice in order to satisfy the existence and stability constraints.

Here, we map the same constraints from *Equation 5* onto bisubstrate autocatalytic reactions. This mapping results in constraints on the assimilated metabolite concentration, as well as on the kinetic parameters of the bisubstrate autocatalytic reactions. We show that in all ternary enzyme complex bisubstrate reaction schemes, there is a lower bound on the concentration of the assimilated metabolite, below which the system cannot attain a stable positive steady state. We further show that the nature of the bisubstrate reaction qualitatively affects the ability to satisfy the stability constraints by changing expression levels alone. In the cases of substituted enzyme mechanism, random binding order ternary complex, and ordered binding ternary complex, with the assimilated metabolite binding first, unless the kinetic parameters of the participating enzymes satisfy specific inequalities, changes to the maximal reaction rates alone cannot suffice in order to satisfy the existence and stability constraints. However, in the case of ordered binding ternary complex with the internal metabolite binding first, changes to the maximal reaction rates alone suffice in order to allow for stable steady state to occur, given high enough concentration of the assimilated metabolite. We analyze each of the four possible bisubstrate reaction schemes separately below.

### Substituted enzyme (Ping Pong) mechanism

The case of substituted enzyme mechanism is the simplest case to analyze. We can substitute *Equation 9* into the conditions from *Equation 5* to get:

$$\begin{cases} V_{\max,b} > \frac{V_{\max}A}{K_A+A} = V_{\max}\frac{A}{K_A+A} \\ \frac{V_{\max,b}}{K_{M,b}} < \frac{V_{\max}}{K_X} \end{cases} \tag{16}$$

As $\frac{A}{K_A+A} < 1$, the first inequality can always be satisfied if $V_{\max,b} > V_{\max}$, which is equivalent to the first condition in *Equation 5*. The second condition is identical to the second condition from *Equation 5*. Therefore, this case imposes equivalent conditions to those derived for the simple, single substrate case.

### Random binding order

In the case of a random binding order, we can substitute *Equation 11* into the conditions from *Equation 5* to get:

$$\begin{cases} V_{\max,b} > \frac{V_{\max}A}{K_A+A} = V_{\max}\frac{A}{K_A+A} \\ \frac{V_{\max,b}}{K_{M,b}} < \frac{V_{\max}A}{(K_{i,A}+A)K_X} = \frac{V_{\max}\frac{A}{K_{i,A}+A}}{K_X} \end{cases} \tag{17}$$

We first note that from the second inequality we get that:

$$\frac{V_{\max,b}K_X}{K_{M,b}V_{\max}} < \frac{A}{K_{i,A}+A} \Rightarrow \frac{K_{i,A}}{\frac{K_{M,b}V_{\max}}{V_{\max,b}K_X}-1} < A \tag{18}$$

Giving a lower bound on the concentration of the assimilated metabolite for which a stable steady state is attainable.

We now wish to obtain a lower bound on $K_{M,b}$. In order to obtain such a lower bound, we need an upper bound on $V_{\max}\frac{A}{K_{i,A}+A}$. However, we only have an upper bound on $V_{\max}\frac{A}{K_A+A}$. We use the first inequality in *Equation 17* to get that:

$$\begin{aligned} V_{\max,b} &> V_{\max}\frac{A}{K_A+A} \Rightarrow \\ V_{\max,b}\frac{A}{K_{i,A}+A} &> V_{\max}\frac{A}{K_A+A}\frac{A}{K_{i,A}+A} \Rightarrow \\ V_{\max,b}\frac{K_A+A}{K_{i,A}+A} &> V_{\max}\frac{A}{K_{i,A}+A} \end{aligned}$$

We note that for positive $A$, $\frac{K_A+A}{K_{i,A}+A} < \max(1,K_A/K_{i,A})$ and therefore:

$$V_{\max,b}\max(1,K_A/K_{i,A}) > V_{\max}\frac{A}{K_{i,A}+A}$$

Substituting this inequality in the second inequality of *Equation 17* therefore gives us that:

$$\begin{aligned} \frac{V_{\max,b}}{K_{M,b}} &< \frac{V_{\max}\frac{A}{K_{i,A}+A}}{K_X} < \frac{V_{\max,b}\max(1,K_A/K_{i,A})}{K_X} \Rightarrow \\ \frac{K_X}{\max(1,K_A/K_{i,A})} &< K_{M,b} \end{aligned} \tag{19}$$

We have therefore obtained a lower bound on the affinity of the branch reaction, $K_{M,b}$, in this case.

For the random binding order we can thus conclude that, like in the single-substrate case, a lower bound exists on the affinity of the branch reaction, below which a positive steady state is not attainable, even if the expression levels of the enzymes, and the concentration of the assimilated metabolite are modified. Furthermore, for any set of kinetic parameters, there is a lower bound on the concentration of $A$ for which a positive steady state is attainable.

## Ordered binding with the assimilated metabolite binding first

In the case of ordered binding, with the assimilated metabolite binding first, we can substitute *Equation 13* into the conditions from *Equation 5* to get:

$$\begin{cases} V_{\max,b} > V_{\max} \\ \frac{V_{\max,b}}{K_{M,b}} < \frac{V_{\max}A}{(K_{i,A}+A)K_X} = \frac{V_{\max}\frac{A}{K_{i,A}+A}}{K_X} \end{cases} \tag{20}$$

As the second inequality is identical to the one in the random binding order case, we can immediately conclude that the same lower bound on the concentration of $A$ from *Equation 18* holds in this case as well.

Regarding a lower bound on $K_{M,b}$, following a similar reasoning as in the previous case, we first note that for any value of $A$:

$$\frac{A}{K_{i,A}+A} < 1$$

Therefore, we can deduce, by using the first inequality from *Equation 20* in the second inequality from that equation, that:

$$\frac{V_{\max,b}}{K_{M,b}} < \frac{V_{\max}\frac{A}{K_{i,A}+A}}{K_X} < \frac{V_{\max}}{K_X} < \frac{V_{\max,b}}{K_X}$$

which immediately yields:

$$K_X < K_{M,b} \tag{21}$$

setting an absolute lower bound on $K_{M,b}$.

We thus arrive at the same conclusions in this case, as we have arrived to in the previous case, namely that a lower bound exists on the affinity of the branch reaction, and that, for any set of kinetic parameters, there is a lower bound on the concentration of the assimilated metabolite, below which a positive stable steady state cannot be obtained.

## Ordered binding with the internal metabolite binding first

In the case of ordered binding, with the internal metabolite binding first, we can substitute *Equation 15* into the conditions from *Equation 5* to get:

$$\begin{cases} V_{\max,b} > \frac{V_{\max}A}{K_A+A} \\ \frac{V_{\max,b}}{K_{M,b}} < \frac{V_{\max}A}{K_{i,X}K_A} \end{cases} \tag{22}$$

As in the previous two cases, the second inequality can be used to obtain a lower bound on the concentration of $A$:

$$\frac{V_{\max,b}K_{i,X}K_A}{K_{M,b}V_{\max}} < A \tag{23}$$

However, unlike in the previous two cases, in this case if $V_{\max} < V_{\max,b}$, then the first inequality in *Equation 22* holds for any concentration of $A$, and, for any concentration of $A$ that is larger than its lower bound, the second inequality is also satisfied, resulting in a stable steady state. This case is therefore more robust than the other cases as it allows for the conditions to be satisfied, at least for high concentrations of $A$, given any set of kinetic parameters.

## Dependence of steady state concentration on assimilated metabolite

*Equation 2* shows the dependency between the steady state concentration of the internal metabolite $X$, $X^*$, and the kinetic parameters of the reactions in the system. Substituting the dependencies of the apparent kinetic parameters from *Equations 11, 13, and 15* into *Equation 2* gives the dependency of $X^*$ on the kinetic parameters of the bisubstrate reactions and the concentration of the assimilated metabolite, $A$. We get for these three cases respectively that:

$$X^* = \frac{\frac{V_{\max}A}{K_A+A}K_{M,b} - V_{\max,b}\frac{K_{i,A}+A}{K_A+A}K_X}{V_{\max,b} - \frac{V_{\max}A}{K_A+A}} \tag{24}$$

$$X^* = \frac{V_{\max}K_{M,b} - V_{\max,b}\frac{K_{i,A}+A}{A}K_X}{V_{\max,b} - V_{\max}} \tag{25}$$

$$X^* = \frac{\frac{V_{\max}A}{K_A+A}K_{M,b} - V_{\max,b}\frac{K_{i,X}K_A}{K_A+A}}{V_{\max,b} - \frac{V_{\max}A}{K_A+A}} \tag{26}$$

Assuming the kinetic parameters satisfy the stable steady state conditions derived in *Equations 17, 20, and 22*, we note that when $A$ is equal to its lower bound, the numerator in all three cases is 0, resulting in $X^* = 0$. Furthermore, as $A$ decreases towards its lower bound, $X^*$ decreases resulting in a decrease in both $f_b$ and $f_a$ (for the two latter cases this is trivial to show, as the terms involving $A$ increase and decrease monotonically in accordance with their effect on $X^*$. In the first case, taking the derivative of the numerator w.r.t. $A$ shows the derivative is always positive, resulting in the same conclusion). Interestingly, in the first and last cases, if $V_{\max} > V_{\max,b}$, then an upper bound

on the concentration of $A$ also exists. As the concentration of $A$ approaches this upper bound, the denominator approaches 0 resulting in an increase in the concentration of $X^*$ towards infinity.

## Reversible branch reaction analysis

The simple model assumed both the autocatalytic and the branch reactions are irreversible. Here we assume the branch reaction is reversible, and let $Y$ denote its product. For simplicity, we further assume that $K_{\mathrm{eq}} = 1$, noting that this assumption can always be satisfied by measuring the concentration of $Y$ in units of $K_{\mathrm{eq}}X$. We recall that the reversible Michaelis-Menten equation states that:

$$f_b = \frac{V_{\mathrm{max},b}(X - Y)}{K_X + X + \frac{K_X}{K_Y}Y}$$

We assume that a third reaction, $f_c$, irreversibly consumes $Y$. While assuming $f_c$ follows irreversible Michaelis-Menten kinetics is analytically tractable, the analysis is simpler, and as informative, under the assumption that $f_c = DY$ for some constant $D$. This simplification is equivalent to assuming $f_c$ follows Michaelis-Menten kinetics with $\frac{V_{\mathrm{max},c}}{K_{M,c}} \approx D$, and $V_{\mathrm{max},c} >> \max(V_{\mathrm{max},a}, V_{\mathrm{max},b})$.

We start by deriving the necessary conditions for steady state existence. Because at steady state $f_a = f_c$, it follows that:

$$\frac{V_{\mathrm{max},a}X^*}{K_{M,a} + X^*} = DY^* \Rightarrow Y^* = \frac{\frac{V_{\mathrm{max},a}}{D}X^*}{K_{M,a} + X^*} \tag{27}$$

Furthermore, as at the steady state $f_a = f_b$, we get that:

$$\frac{V_{\mathrm{max},a}X^*}{K_{M,a} + X^*} = \frac{V_{\mathrm{max},b}(X^* - Y^*)}{K_X + X^* + \frac{K_X}{K_Y}Y^*}$$

Substituting $Y^*$ from *Equation 27* gives:

$$\frac{V_{\mathrm{max},a}X^*}{K_{M,a} + X^*} = \frac{V_{\mathrm{max},b}\left(X^* - \frac{\frac{V_{\mathrm{max},a}}{D}X^*}{K_{M,a} + X^*}\right)}{K_X + X^* + \frac{K_X}{K_Y}\frac{\frac{V_{\mathrm{max},a}}{D}X^*}{K_{M,a} + X^*}}$$

Which is satisfied when $X^* = 0$ (implying that $X^* = Y^* = 0$ is a steady state), or when $X^*$ satisfied the quadratic equation:

$$0 = (X^*)^2 + \frac{2K_{M,a}V_{\mathrm{max},b} - (K_{M,a} + K_X)V_{\mathrm{max},a} - \frac{K_X V_{\mathrm{max},a}^2}{K_Y D} - \frac{V_{\mathrm{max},a}V_{\mathrm{max},b}}{D}}{V_{\mathrm{max},b} - V_{\mathrm{max},a}}X^* +$$
$$\frac{K_{M,a}\left(V_{\mathrm{max},b}K_{M,a} - V_{\mathrm{max},a}K_X - \frac{V_{\mathrm{max},a}V_{\mathrm{max},b}}{D}\right)}{V_{\mathrm{max},b} - V_{\mathrm{max},a}}$$

Albeit intimidating, this quadratic equation can be used to derive the conditions for the existence of a positive steady state. Only if both of the roots of this equation are negative, no positive steady state exists. We recall that the two roots of a quadratic equation of the form $0 = aX^2 + bX + c$ are negative iff:

$$\begin{cases} b &= \frac{2K_{M,a}V_{\mathrm{max},b} - (K_{M,a} + K_X)V_{\mathrm{max},a} - \frac{K_X V_{\mathrm{max},a}^2}{K_Y D} - \frac{V_{\mathrm{max},a}V_{\mathrm{max},b}}{D}}{V_{\mathrm{max},b} - V_{\mathrm{max},a}} > 0 \\ c &= \frac{K_{M,a}\left(V_{\mathrm{max},b}K_{M,a} - V_{\mathrm{max},a}K_X - \frac{V_{\mathrm{max},a}V_{\mathrm{max},b}}{D}\right)}{V_{\mathrm{max},b} - V_{\mathrm{max},a}} > 0 \end{cases}$$

As in the irreversible case, the sign of $V_{\mathrm{max},b} - V_{\mathrm{max},a}$ determines the required condition on the numerators. We assume that $V_{\mathrm{max},b} > V_{\mathrm{max},a}$, noting that if $V_{\mathrm{max},b} < V_{\mathrm{max},a}$, a positive steady state cannot be globally stable because for $X$ such that $f_a(X) > V_{\mathrm{max},b}$, the system will diverge regardless of the value of $Y$.

Under the assumption that $V_{\max,b} > V_{\max,a}$, the denominator of both $b$ and $c$ is positive, meaning a positive steady state exists only if the nominators of $b$ or $c$ (or both) are negative. Thus, two options may arise.

- If $K_{M,a} > V_{\max,a}/D$ (implying $D > V_{\max,a}/K_{M,a}$, qualitatively suggesting rapid removal of $Y$) then an upper bound on $V_{\max,b}$ exists, above which the two solutions are negative, implying no positive steady state exists. A sufficient condition for existence in this case is that $\frac{V_{\max,b}}{K_X} < \frac{V_{\max,a}}{K_{M,a}}$, ensuring that $c < 0$. This condition is equivalent to the condition in the irreversible case. We further show below that for large enough $D$, the resulting steady state is stable.
- If $D < V_{\max,a}/K_{M,a}$, then for any $V_{\max,b} > V_{\max,a}$, $c < 0$ implying a positive steady state exists. As we show below, in this case both when $V_{\max,b} \to V_{\max,a}$, and when $V_{\max,b} \to \infty$, the steady state is stable.

We now turn to analyze the stability of the steady state. For a steady state to be stable, the eigenvalues of the Jacobian matrix must have negative real values. In our system it holds that

$$\dot{X} = f_a - f_b$$
$$\dot{Y} = f_b - f_c$$

We use the following notation:

$$\alpha = \frac{df_a}{dX} = \frac{V_{\max,a} K_{M,a}}{(K_{M,a} + X)^2}$$

$$\beta_x = \frac{\partial f_b}{\partial X} = \frac{V_{\max,b}(K_X + Y(1 + \frac{K_X}{K_Y}))}{(K_X + X + \frac{K_X Y}{K_Y})^2}$$

$$\beta_y = \frac{\partial f_b}{\partial Y} = \frac{-V_{\max,b}(K_X + X(1 + \frac{K_X}{K_Y}))}{(K_X + X + \frac{K_X Y}{K_Y})^2}$$

$$\frac{df_c}{dY} = D$$

We can use this notation to write the Jacobian matrix as:

$$J = \begin{pmatrix} \alpha - \beta_x & -\beta_y \\ \beta_x & \beta_y - D \end{pmatrix}$$

which gives a characteristic polynomial of:

$$(\alpha - \beta_x - \lambda)(\beta_y - D - \lambda) + \beta_y \beta_x = 0$$

In order for the real values of the roots of the characteristic polynomial to be negative it must hold that $b > 0$ and $c > 0$, where $b$ and $c$ are now the coefficients of the quadratic equation $a\lambda^2 + b\lambda + c = 0$. We therefore get that:

$$\begin{cases} b = \beta_x - \alpha - \beta_y + D > 0 \\ c = (\alpha - \beta_x)(\beta_y - D) + \beta_y \beta_x = \beta_x D + \alpha \beta_y - \alpha D > 0 \end{cases}$$

We denote by $f^*$ the steady state flux in the system, such that $f^* = f_a = f_b = f_c$ We note that for MM kinetics and positive concentrations it holds that:

$$\alpha > 0$$
$$\beta_x > 0$$
$$-\beta_y > \beta_x$$
$$\beta_x + \beta_y = -f^* \frac{1 + \frac{K_X}{K_Y}}{K_X + X + \frac{K_X Y}{K_Y}}$$

First, we note that if $\alpha \geq \beta_x$ then the steady state cannot be stable as, looking at the value of $c$, we see that in such a case $(\beta_x - \alpha)D < 0$ and since $\alpha\beta_y < 0$, $c < 0$ violating the stability conditions. However, because we assume that $V_{\max,b} > V_{\max,a}$, then for $Y = Y^*$, at $X = 0$, $f_b < f_a$, but for $X \to \infty$, $f_b \to V_{\max,b}$ and $f_a \to V_{\max,a}$, so that $f_b > f_a$. It then follows that, because the two fluxes can only intersect

once for positive $X$ and fixed $Y$, at the steady state point, where $f_a = f_b$, $\alpha < \beta_x$, so this condition is satisfied. We note that this condition is sufficient to ensure that $b > 0$. We also note that as $\alpha < \beta_x$, a large enough value of $D$ exists at which the steady state is stable, concluding that if $D$ is large enough, then a stable steady state exists if:

$$\begin{cases} V_{\max,b} & > V_{\max,a} \\ \frac{V_{\max,b}}{K_X} & < \frac{V_{\max,a}}{K_{M,a}} \end{cases}$$

If $D$ is small, such that $D < V_{\max,a}/K_{M,a}$, and $V_{\max,b} > V_{\max,a}$ (implying that $\alpha < \beta_x$), we need to check what other conditions are necessary in order to ensure that $\beta_x D + \alpha \beta_y - \alpha D > 0$. We look at the limit $V_{\max,b} \to \infty$. At this limit, the quadratic equation for $X^*$ converges to:

$$0 = (X^*)^2 + \left( 2K_{M,a} - \frac{V_{\max,a}}{D} \right) X^* + K_{M,a}^2 - \frac{V_{\max,a} K_{M,a}}{D}$$

For this equation, $c < 0$, implying that one of the roots is negative and one is positive. The positive root is:

$$X^* = \frac{V_{\max,a}}{D} - K_{M,a}$$

As this $X^*$ is finite, we get that when $V_{\max,b} \to \infty$, $Y^*$ also converges to $\frac{V_{\max,a}}{D} - K_{M,a}$. At this limit, $\beta_x$ increases infinitely and $\beta_y$ decreases infinitely, but $\beta_x + \beta_y$ converges to:

$$-f^* \frac{\left(1 + \frac{K_X}{K_Y}\right)}{K_X + \left(\frac{V_{\max,a}}{D} - K_{M,a}\right)\left(1 + \frac{K_X}{K_Y}\right)}$$

that is constant. Therefore, rearranging c such that:

$$c = (\beta_x + \beta_y)D - \beta_y(D - \alpha) - \alpha D > 0$$

we note that as $V_{\max,b}$ increases, the dominant term becomes $-\beta_y(D - \alpha) > 0$ ensuring that $c > 0$ and therefore stability.

On the other hand, when $V_{\max,b} \to V_{\max,a}$, we note that because $f_c < V_{\max,a}$, $Y^*$ is bounded by $Y^* < \frac{V_{\max,a}}{D}$, but $X^* \to \infty$. Thus, both $\alpha$ and $\beta_x$ diminish like $\frac{1}{X^*}^2$, and $\beta_y$ diminishes like $\frac{1}{X^*}$. The dominant term in $c = \beta_x D + \beta_y \alpha - \alpha D$ therefore becomes $(\beta_x - \alpha)D > 0$ so again stability is maintained.

Therefore, for small values of $D$, as long as $V_{\max,b} > V_{\max,a}$, a positive stable steady state exists both in the lower limit of $V_{\max,b} \to V_{\max,a}$, and in the upper limit of $V_{\max,b} \to \infty$.

Our conclusions are therefore as follows: As in the irreversible case, $V_{\max,b} > V_{\max,a}$ is a necessary condition for the existence of a globally stable steady state. For large values of $D$, the reversible reaction is far from equilibrium, resulting in an additional condition, equivalent to the condition we obtained for the irreversible case, namely that $V_{\max,b}/K_X$ is upper bounded by a term that is larger than $V_{\max,a}/K_{M,a}$, but approaches it as $D$ increases. This condition is sufficient for the existence and stability of the steady state. For small values of $D$, a steady state always exists (given that $V_{\max,b} > V_{\max,a}$). We can show that this steady state is stable both when $V_{\max,b} \to \infty$, and when $V_{\max,b} \to V_{\max,a}$. We therefore conclude that in this case, no further restrictions apply on $K_X$, $K_Y$, or $K_{M,a}$ but rather that a steady state can always be achieved at most by changing $V_{\max,b}$.

Qualitatively, the cases we analyze show that, on top of the required $V_{\max,b} > V_{\max,a}$ condition, the second condition is that either the slope of $f_c = D$ is smaller than $V_{\max,a}/K_{M,a}$, or that the maximal slope of $f_b$, $V_{\max,b}/K_X$, is smaller than $V_{\max,a}/K_{M,a}$.

## Extending the stability analysis from single to multiple reaction cycles

We analyze the stability criteria for the autocatalytic cycles depicted in *Figure 5A and B*. We start by writing the relevant equations for the autocatalytic cycle depicted in *Figure 5A*. In this system, there are two intermediate metabolites, $X_1$ and $X_2$, two reactions that form the cycle, $f_{a_1}$ and $f_{a_2}$, and two branch reactions, $f_{b_1}$ and $f_{b_2}$. We assume, without loss of generality, that the autocatalytic reaction

(the reaction that increases the amount of carbon in the cycle) is $f_{a_2}$ and that the autocatalysis is in a 1:2 ratio. The equations describing the dynamics of the system are thus:

$$\dot{X}_1 = 2f_{a_2} - f_{a_1} - f_{b_1}$$
$$\dot{X}_2 = f_{a_1} - f_{a_2} - f_{b_2}$$

We note that in steady state, where $\dot{X}1 = \dot{X}2 = 0$, because the autocatalysis is in a 1:2 ratio, it must hold that $f_{b_1} + f_{b_2} = f_{a_2}$, meaning the total outgoing flux balances the total increase of intermediate metabolites due to autocatalysis. Given that a steady state of the system exists for some value $(X_1^*, X_2^*)$, we can evaluate the condition for stability. In multi-variable systems, stability dictates that the real part of the eigenvalues of the Jacobian matrix must all be negative. We define $\alpha_i = \frac{\partial f_{a_i}}{\partial X_i}$ and $\beta_i = \frac{\partial f_{b_i}}{\partial X_i}$ for $i = 1, 2$. We note that as we assume Michaelis Menten kinetics, $\alpha_i > 0$ and $\beta_i \geq 0$, where $\beta_i = 0$ is the case where there is no flux branching out at $i$. We then get that the Jacobian matrix is:

$$J = \begin{pmatrix} -(\alpha_1 + \beta_1) & 2\alpha_2 \\ \alpha_1 & -(\alpha_2 + \beta_2) \end{pmatrix}$$

Solving for the characteristic polynomial gives:

$$0 = (\lambda + \alpha_1 + \beta_1)(\lambda + \alpha_2 + \beta_2) - 2\alpha_1\alpha_2$$
$$= \lambda^2 + (\alpha_1 + \beta_1 + \alpha_2 + \beta_2)\lambda + (\alpha_1 + \beta_1)(\alpha_2 + \beta_2) - 2\alpha_1\alpha_2$$

that has two negative roots when:

$$(\alpha_1 + \beta_1)(\alpha_2 + \beta_2) - 2\alpha_1\alpha_2 > 0 \Rightarrow (1 + \frac{\beta_1}{\alpha_1})(1 + \frac{\beta_2}{\alpha_2}) > 2$$

which is satisfied if $\beta_1 > \alpha_1$ or $\beta_2 > \alpha_2$. Therefore, if either $\beta_1 > \alpha_1$ or $\beta_2 > \alpha_2$ at the steady state, then the steady state is stable.

The two-metabolites cycle case can be easily extended to a larger number of intermediate metabolites and reactions, as is depicted in **Figure 5B**. For this extension, we again assume, without loss of generality, that the autocatalytic reaction is the last reaction, $f_{a_n}$, and that the autocatalysis is in a 1:2 ratio.

In this case, steady state implies that the concentration of each intermediate metabolite is conserved, meaning that for all $i > 1$:

$$\dot{X}_i = 0 \Rightarrow f_{a_{i-1}} - f_{a_i} - f_{b_i} = 0 \Rightarrow f_{a_{i-1}} \geq f_{a_i} \tag{28}$$

(for $i = 1$, as $f_{a_n}$ is the autocatalytic reaction, we get that $2 \cdot f_{a_n} \geq f_{a_1}$). Also, because at steady state the total outgoing flux from the cycle must balance the total incoming flux into the system, which is the amount of autocatalysis carried out by $f_{a_n}$, we get that:

$$\sum_{i=1}^{n} f_{b_i} = f_{a_n}$$

(due to our assumption of a 1:2 autocatalytic ratio) implying that for all $i$:

$$f_{b_i} \leq f_{a_n} \tag{29}$$

We stress that **Equation 29** is only valid if the autocatalysis is in up to a 1:2 ratio. Deriving a stability criterion for the multiple-reaction case, we get that in this case a steady state is stable if there exists $i$ such that $\beta_i > \alpha_i$ (see section 9 below).

To conclude, for the straightforward extension of the simple model to multiple reactions with a single autocatalytic reaction, steady state implies that for all $i$:

$$f_{b_i} \leq f_{a_n} \leq f_{a_i} \tag{30}$$

Where the left inequality is due to **Equation 29** and the right inequality is due to **Equation 28**.

A sufficient condition for such a steady state point to be stable is that at the steady state point there exists at least one branching point $i$ at which the derivative of the branch reaction is larger than the derivative of the equivalent autocatalytic reaction:

$$\beta_i > \alpha_i \tag{31}$$

## Limits on derivatives of branch reactions for complex autocatalytic cycles

Stability analysis of a model complex autocatalytic cycle with $n$ reactions in the cycle results in the following Jacobian matrix:

$$J = \begin{pmatrix} -(\alpha_1+\beta_1) & 0 & \cdots & 0 & 2\alpha_n \\ \alpha_1 & -(\alpha_2+\beta_2) & \cdots & 0 & 0 \\ \vdots & \vdots & \ddots & \vdots & \vdots \\ 0 & 0 & \cdots & -(\alpha_{n-1}+\beta_{n-1}) & 0 \\ 0 & 0 & \cdots & \alpha_{n-1} & -(\alpha_n+\beta_n) \end{pmatrix} \tag{32}$$

The characteristic polynomial of this matrix is given by:

$$0 = \prod_{i=1}^{n}(\lambda+\alpha_i+\beta_i) - 2\prod_{i=1}^{n}\alpha_i \tag{33}$$

To extract the conditions under which all the roots of the characteristic polynomial have negative real parts we use Rouche's theorem. Our strategy will be as follows: We will define a contour that contains only numbers with negative real parts. We will show that all the roots of the polynomial $0 = \prod_{i=1}^{n}(\lambda+\alpha_i+\beta_i)$ lie within the area this contour encloses. We will find the conditions for which $|\prod_{i=1}^{n}(\lambda+\alpha_i+\beta_i)| > 2\prod_{i=1}^{n}\alpha_i$ on the contour, satisfying the premise of Rouche's theorem. We will then claim that under these conditions all the roots of the polynomial in **Equation 33** must also lie inside the contour, and therefore must have negative real parts. Given that all the roots of this polynomial have negative real parts, we will conclude that the eigenvalues of the Jacobian matrix at the steady state point all have negative real parts, making the steady state point stable.

### Proof

We pick a large parameter $R$, such that $R > 3 \max_j(\alpha_j + \beta_j)$. We look at the closed half circle contour, $K$, composed of the segment $[(0,-iR),(0,iR)]$ and the half circle arc $(x, iy)$ such that $x \leq 0$ and $x^2 + y^2 = R^2$. We define

$$g(\lambda) = 2\prod_{j=1}^{n}\alpha_j$$

noting that it is constant over all of $\mathbb{C}$ and specifically over $K$. We define

$$f(\lambda) = \prod_{j=1}^{n}(\lambda+\alpha_j+\beta_j)$$

noting that all of $f$'s roots lie inside $K$ as the roots are $0 > -(\alpha_j+\beta_j) > -R$ for all $j$. We check the conditions under which $|f(\lambda)| > |g(\lambda)|$ over the contour $K$.

For the arc segment we note that, as for complex numbers it holds that $|xy| = |x||y|$, then

$$|f(\lambda)| = \prod_{j=1}^{n}|\lambda+\alpha_j+\beta_j|$$

From the triangle inequality we know that $|x+y| \geq |x| - |y|$ and therefore for all $j$ it holds that

$$|\lambda+\alpha_j+\beta_j| \geq R - (\alpha_j+\beta_j)$$

As we picked $R$ such that $R > 3 \max_j(\alpha_j+\beta_j)$ we get that

$$R - (\alpha_j + \beta_j) > 2(\alpha_j + \beta_j)$$

and therefore

$$\prod_{j=1}^{n} |\lambda + \alpha_j + \beta_j| > \prod_{j=1}^{n} 2|\alpha_j + \beta_j| > \prod_{j=1}^{n} 2|\alpha_j| = |g(\lambda)|$$

concluding that over the arc, $|f(\lambda)| > |g(\lambda)|$.

For the part of $K$ on the imaginary axis, we note that $\lambda = iy$ where $y \in [-R, R]$. For this segment we therefore get that

$$|f(\lambda)| = \prod_{j=1}^{n} |\alpha_j + \beta_j + iy| = \sqrt{\prod_{j=1}^{n} ((\alpha_j + \beta_j)^2 + y^2)} \geq \sqrt{\prod_{j=1}^{n} (\alpha_j + \beta_j)^2}$$

and, as before, that

$$|g(\lambda)| = 2\sqrt{\prod_{j=1}^{n} \alpha_j^2}$$

To meet the condition that $|f(\lambda)| > |g(\lambda)|$, which is equivalent to: $\frac{|f(\lambda)|}{|g(\lambda)|} > 1$, it is sufficient to find the conditions under which:

$$\frac{\sqrt{\prod_{j=1}^{n} (\alpha_j + \beta_j)^2}}{2\sqrt{\prod_{j=1}^{n} \alpha_j^2}} > 1$$

Simplifying this inequality gives:

$$\frac{1}{2}\sqrt{\prod_{j=1}^{n} \frac{(\alpha_j + \beta_j)^2}{\alpha_j^2}} = \frac{1}{2}\prod_{j=1}^{n} \frac{\alpha_j + \beta_j}{\alpha_j} = \frac{1}{2}\prod_{j=1}^{n} (1 + \frac{\beta_j}{\alpha_j}) > 1 \Rightarrow \prod_{j=1}^{n} (1 + \frac{\beta_j}{\alpha_j}) > 2$$

A sufficient condition to satisfy this inequality, given that all the $\alpha_j$ are positive and all the $\beta_j$'s are non negative, is that there exists $j$ such that $\beta_j > \alpha_j$.

We therefore get that if there exists $j$ such that $\beta_j > \alpha_j$, then $|f(\lambda)| > |g(\lambda)|$ over the contour $K$. In this case, by Rouche's theorem, we deduce that, as all of $f$'s roots lie inside $K$, then it follows that all of $f - g$'s roots lie inside $K$, concluding that the real part of all of the eigenvalues of the characteristic polynomial of the Jacobian matrix of the complex autocatalytic cycle are negative, making any steady state that meets this criterion stable.

## Multiple unsaturated branch reactions increase convergence speed and dampen oscillations

Using the Jacobian matrix from *Equation 32* we can analyze the effect of multiple low saturation branch points on convergence to steady state. The analysis shows that the more $i$'s exist for which $\beta_i > 0$, and the larger $\beta_i$ is (resulting in lower saturation of $f_{b_i}$), the faster the convergence of the cycle to steady state will be.

We denote by $\vec{X}^*$ the steady state vector of the concentrations of the intermediate metabolites. We denote by $\vec{X} = \vec{X}^* + \Delta X_j$ a state where for all the intermediate metabolites that are not $X_j$, their concentration is the same as the steady state concentration, and $X_j$ differs by a small amount, $\Delta X_j$, from its steady state concentration. We let $F$ denote the fluxes function of the system such that $F(X) = \dot{X}|_{\vec{x}}$. Evaluating the dynamics of the system at $\vec{X}$ by noting that $F(\vec{X}) \approx F(\vec{X}^*) + J \cdot \Delta X_j = J \cdot \Delta X_j$ (where $F(\vec{X}^*) = 0$ as $\vec{X}^*$ is a steady state) results in $F(\vec{X})_k = 0$ for all $k \neq j, j+1$. For $X_j$ such that $j \neq n$ we get $F(\vec{X})_j \approx -(\alpha_j + \beta_j)\Delta X_j$ and for $X_{j+1}$ we get $F(\vec{X})_{j+1} \approx \alpha_j \Delta X_j$. Therefore, the difference from the steady state decreases proportionally to $\beta_j$ (and cycles to the next intermediate metabolite, $X_{j+1}$). For $j = n$, we get that $F(\vec{X})_j \approx -(\alpha_j + \beta_j)\Delta X_j$, as for $j \neq n$, but $F(\vec{X})_1 \approx 2\alpha_j \Delta X_j$ where the factor of 2 is

due to the effect of the assimilating reaction, that causes an amplification of the deviation from steady state (an amplification that is dampened by subsequent reactions along the cycle if the conditions for stable steady state are satisfied).

It therefore follows that any increase in $\beta_j$, for any $j$, increases the speed of convergence to steady state and reduces the propagation of deviations from steady state for $X_j$. Because of the linearity of matrix multiplication, an arbitrary deviation from $\vec{X^*}$ can always be decomposed to individual deviations with respect to every intermediate metabolite, making the analysis above valid for such deviations as well. Thus, to keep deviations from steady state at check, it is beneficial to increase $\beta_j$, for all $j$, which implies decreasing the saturation of $f_{b_j}$.

## Inverse relationship between derivatives, affinities, and saturation levels

It turns out that for the Michaelis-Menten kinetics equations, the following useful lemma can be used to connect theoretical observations on the relationships of derivatives to physiological observations on affinities and saturation levels.

We define the saturation level of a reaction as the ratio between the flux it carries, and the maximal flux it can carry, given the expression level of the relevant enzyme, that is:

$$S(X) = \frac{f(X)}{V_{\max}} = \frac{X}{K_M + X}$$

Given this definition we can show that if two Michaelis-Menten reactions consume the same metabolite, $X$, and at a given concentration, $X^*$, it holds that $f_a(X^*) \geq f_b(X^*)$, then if:

$$\left.\frac{df_b}{dX}\right|_{X=X^*} > \left.\frac{df_a}{dX}\right|_{X=X^*} \tag{34}$$

then it follows that:

$$\begin{cases} K_{M,b} > K_{M,a} \\ S_b(X^*) < S_a(X^*) \end{cases} \tag{35}$$

Proof: expanding the condition that $f_a(X^*) \geq f_b(X^*)$, we get that:

$$\frac{V_{\max,b}X^*}{K_{M,b} + X^*} \leq \frac{V_{\max,a}X^*}{K_{M,a} + X^*} \Rightarrow \frac{V_{\max,b}}{K_{M,b} + X^*} \leq \frac{V_{\max,a}}{K_{M,a} + X^*} \tag{36}$$

Expanding the premise of the lemma in **Equation 34** gives us that:

$$\left.\frac{df_b}{dX}\right|_{X=X^*} > \left.\frac{df_a}{dX}\right|_{X=X^*} \Rightarrow \frac{V_{\max,b}K_{M,b}}{(K_{M,b} + X^*)^2} > \frac{V_{\max,a}K_{M,a}}{(K_{M,a} + X^*)^2}$$

Because **Equation 36** holds, it follows that:

$$\frac{K_{M,b}}{K_{M,b} + X^*} > \frac{K_{M,a}}{K_{M,a} + X^*} \Rightarrow \frac{1}{1 + \frac{X^*}{K_{M,b}}} > \frac{1}{1 + \frac{X^*}{K_{M,a}}} \Rightarrow K_{M,b} > K_{M,a}$$

setting the affinity of the autocatalytic enzyme as a lower bound for the affinity of the branch enzyme. Finally, given this relation of affinities it follows that:

$$K_{M,b} > K_{M,a} \Rightarrow X^* + K_{M,b} > X^* + K_{M,a} \Rightarrow \frac{X^*}{X^* + K_{M,b}} < \frac{X^*}{X^* + K_{M,a}} \Rightarrow S_b(X^*) < S_a(X^*)$$

concluding the proof.

We note that a multiple reaction autocatalytic cycle at a stable steady state point satisfies **Equations 30 and 31**, so the lemma applies.

## Evaluating maximal flux capacity of reactions under a given condition

To evaluate the maximal flux capacity of a reaction under a prescribed growth condition, given expression level and flux data for a set of conditions, we follow the procedure described in

*Davidi et al. (2016)*. For each reaction, under every condition, we divide the flux the reaction carries (obtained from *Gerosa et al., 2015*) by the amount of the corresponding enzyme expressed under that condition (obtained from *Schmidt et al., 2016*). We thus get a flux per enzyme estimate for the given reaction under each of the conditions. We define the enzyme maximal in-vivo catalytic rate as the maximum flux per unit enzyme it carries across all conditions analyzed (noting that this is actually only a lower bound on this rate). Multiplying the enzyme maximal catalytic rate by the enzyme amount at each condition results in an estimate of the maximal possible flux through the given reaction under the relevant condition.

## Allosteric regulation can improve network performance

In this section we touch upon the potential (in somewhat simplified and naively non-rigorous terms) of allosteric regulation to improve the properties of autocatalytic cycles. The constraint on the affinity of the branch reaction imposed by the stability requirement (*Equation 35*) may be suboptimal under other flux modes. Furthermore, allosteric regulation can be used to accelerate the rate at which an autocatalytic cycle converges to its stable steady state mode. While many allosteric regulation schemes exist (*Leskovac, 2003*), all of these schemes affect the affinity of the regulated enzyme, and some of these schemes also affect the maximal rate. We qualitatively analyze the expected regulation benefits for autocatalytic cycles.

From the perspective of the simple model, we recall that $\dot{X} = f_a - f_b$. If the cycle is such that some steady state concentration, $X^*$, is the desired value for biological function, then for levels of $X$ below $X^*$ convergence will be faster if $f_a$ is increased and $f_b$ is decreased, compared with their values at $X^*$. Conversely, for levels of $X$ above $X^*$, convergence will be faster if $f_a$ is decreased and $f_b$ is increased, compared with their values at $X^*$. Convergence to $X^*$ can therefore be accelerated if, for example, $X$ activates the branch reactions and inhibits the cycle reactions.

The assimilated metabolite can also allosterically regulate the reactions of the cycle. We assume that the desired steady state, denoted $\hat{X}$, does not depend on the concentration of the assimilated metabolite, $A$. Under this assumption, we further assume that $\hat{X}$ is attained for some constant concentration of the assimilated metabolite, $\hat{A}$. It then follows that because the autocatalytic activity is higher when $A > \hat{A}$, then in order to maintain $X^*$ close to its desired level, when $A > \hat{A}$, $f_a$ should be inhibited, and $f_b$ should be activated, but when $A < \hat{A}$, $f_a$ should be activated, and $f_b$ should be inhibited. Therefore, to increase the robustness of the steady state concentration to changes in the concentration of the assimilated metabolite, the assimilated metabolite should inhibit the cycle reactions and activate the branch reactions.

Another possible class of regulators are the products of the branch reactions. Taking a somewhat simplistic view, if the level of $Y$, the product of a branch reaction is low, this can indicate that the cycle does not carry sufficient flux to supply the demand for $Y$. Regulation can then be used to increase $X^*$. From *Equation 7*, we get that the steady state concentration, $X^*$, increases as $K_{M,b}$ increases and $V_{\max,b}$ decreases, corresponding to inhibition of $f_b$, and that $X^*$ decreases as $K_{M,a}$ increases and $V_{\max,a}$ decreases, corresponding to inhibition of $f_a$. So, to tune autocatalytic fluxes to match the demands of $Y$, regulation should increase $X^*$ when $Y$ is low, by activating the recycling and autocatalytic reactions and inhibiting the branch reactions. On the other hand, regulation should decrease $X^*$ when $Y$ is high, by inhibiting the autocatalytic reactions and activating the branch reactions. Therefore, to synchronize the demand of the cycle product with the cycle flux, the cycle branch products should inhibit the cycle reactions and activate the branch reactions.

Finally, we note that in the autocatalytic cycles we identify in central carbon metabolism, there are also reactions that operate in the reverse direction to the branch reactions, such that they consume products of the cycle and produce intermediate metabolites of the cycle. As such reactions are mirror images of branch reactions, we expect them to be oppositely regulated to branch reactions.

We find that these predictions hold for the cycle using the PTS, that is known to be allosterically controlled, but not for the glyoxylate cycle, which is known to be transcriptionally controlled (*Gerosa et al., 2015*).

## Acknowledgements

We would like to thank Arren Bar-Even, Katja Tummler, Matthias Heinemann, David Fell, Patrick Shih, Noam Prywes, Leeat Keren, David Wernick, Wolfram Liebermeister, Rami Pugatch and Ron Sender for fruitful discussions and valuable insights contributing to this work. We would also like to thank Frank Bruggeman and the other anonymous reviewers for their significant contributions to the completeness and coherence of this work.

## Additional information

### Funding

| Funder | Grant reference number | Author |
| --- | --- | --- |
| Israel Science Foundation | 740/16 | Uri Barenholz<br>Dan Davidi<br>Yinon Bar-On<br>Niv Antonovsky<br>Ron Milo |
| Beck-Canadian Center for Alternative Energy Research | | Uri Barenholz<br>Dan Davidi<br>Yinon Bar-On<br>Niv Antonovsky<br>Ron Milo |
| Dana and Yossie Hollander | | Uri Barenholz<br>Dan Davidi<br>Yinon Bar-On<br>Niv Antonovsky<br>Ron Milo |
| Leona M. and Harry B. Helmsley Charitable Trust | | Uri Barenholz<br>Dan Davidi<br>Yinon Bar-On<br>Niv Antonovsky<br>Ron Milo |
| The Larson Charitable Foundation | | Uri Barenholz<br>Dan Davidi<br>Yinon Bar-On<br>Niv Antonovsky<br>Ron Milo |
| Wolfson Family Charitable Trust | | Uri Barenholz<br>Dan Davidi<br>Yinon Bar-On<br>Niv Antonovsky<br>Ron Milo |
| Charles Rothchild | | Uri Barenholz<br>Dan Davidi<br>Yinon Bar-On<br>Niv Antonovsky<br>Ron Milo |
| Selmo Nussenbaum | | Uri Barenholz<br>Dan Davidi<br>Yinon Bar-On<br>Niv Antonovsky<br>Ron Milo |
| Alternative Sustainable Energy Research Initiative | Graduate Student Fellowship | Uri Barenholz |
| European Research Council | NOVCARBFIX 646827 | Uri Barenholz<br>Dan Davidi<br>Yinon Bar-On<br>Niv Antonovsky<br>Ron Milo |

The funders had no role in study design, data collection and interpretation, or the decision to submit the work for publication.

## Author contributions
UB, Conceptualization, Software, Formal analysis, Investigation, Visualization, Writing—original draft; DD, Data curation, Software, Investigation, Visualization, Methodology, Writing—review and editing; ER, Formal analysis, Investigation, Methodology, Writing—review and editing; YB-O, Data curation, Software; NA, Visualization, Methodology, Writing—review and editing; EN, Formal analysis, Validation, Investigation, Visualization, Methodology, Writing—review and editing; RM, Conceptualization, Formal analysis, Supervision, Funding acquisition, Validation, Investigation, Visualization, Methodology, Project administration, Writing—review and editing

## Author ORCIDs
Uri Barenholz, http://orcid.org/0000-0002-3097-9681
Ed Reznik, http://orcid.org/0000-0002-6511-5947
Ron Milo, http://orcid.org/0000-0003-1641-2299

# Additional files

## Supplementary files
• Supplementary file 1. Contains the tables used in the data analysis in this work. The 'contents' sheet includes the description of the different tables and is provided here as well: Fluxes source: The metabolic fluxes sheet from Data S1 in *Gerosa et al. (2016)*. Cell size source: The cell sizes used for calculations as taken from Supplementary tables, Table 'Content and abbreviations' in *Schmidt et al. (2016)*. Protein abundance: Protein abundance data from Supplementary tables, Table S6 in *Schmidt et al. (2016)*. Reaction-Protein mapping: Mapping between reactions and genes of catalyzing enzymes. lux per enzyme: Calculation of the flux per enzyme for all the reactions listed in the 'Reaction-Protein mapping' table. Reaction Saturation: Estimated saturation of enzymes across conditions. Non autocatalytic cycles reaction saturation: Comparison of saturation levels of branch versus cycle reactions for non-autocatalytic cycles. Allosteric regulation: Listing allosteric interactions between autocatalytic components.

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
