## [Decision Letter]

Thank you for submitting your article "Design principles of autocatalytic cycles constrain enzyme kinetics and force over-expression at flux branch points" for consideration by *eLife*. Your article has been reviewed by four peer reviewers, and the evaluation has been overseen by a Reviewing Editor and Aviv Regev as the Senior Editor. The following individuals involved in review of your submission have agreed to reveal their identity: Frank Bruggeman (Reviewer #2).

The reviewers have discussed the reviews with one another and the Reviewing Editor has drafted this decision to help you prepare a revised submission.

Summary:

This article addresses a very interesting and fundamental question about the nature of autocatalytic cycles within cellular metabolic networks, namely whether the existence and stability of steady states in such cycles imposes constraints on the kinetic parameters of the reactions involved. Starting from simple reactions, and gradually looking at slightly more complicated cases, the authors identify general rules that enzymes should follow in order for the autocatalytic cycles to support a stable flux.

All four reviewers agreed that the article is well written and original, and that it provides insightful and valuable results that would be in principle worth presenting to the audience of *eLife*. However, they also requested a number of important clarifications and revisions, as listed cumulatively below:

Essential revisions:

1) In the equation for "fa", there is no mention of the bisubstrate nature of the reaction. It is understood that the authors consider A constant, which seems a reasonable assumption to start with (though it may be interesting to entertain simple alternatives, e.g. as a constant influx of A – see also next point). However, it is not completely obvious that, even if the kinetic parameters relative to A in the M.M. equations are incorporated into the constants of the equation as shown, the bisubstrate nature of the reaction would not affect the generality of the conclusions drawn throughout the paper.

2) Both in the simple model, and in the extended ones, the authors consider A to be constant. This can be a convenient assumption for the simple model, but makes conclusions for realistic models less relevant, since consumption of A (lowering its level to a critical amount) can be an important part of the system's dynamics. The authors should either explore this alternative possibility, or provide a justification for why this would not be expected to affect significantly the main conclusions.

3) Even though the simple example that is treated in the paper nicely sets the stage and helps the reader in understanding the analytical approach, it is based on irreversible enzyme kinetics only. How do the requirements for proper functioning change when the reactions are catalysed by product-sensitive and reversible kinetics? Does product sensitivity prevent loss of stability? How does the displacement from thermodynamic equilibrium of the enzymes at the branch point influence the stability?

4) Allosteric regulation, within or onto, the autocatalytic subnetwork can also improve the robust functioning of the autocatalytic subnetwork. Have the authors analysed this? What did they find? This is particularly relevant in relation to the remark, which suggests that enzyme expression adjustment -- so, in fact rate adjustment -- can always lead to a non-zero, stable steady state. If so, then is it true that allosteric regulation could always lead to a steady state? Could one then make predictions about the kinetic design of autocatalytic cycles based on their allosteric control?

5) In subsection “Stability analysis for multiple-reaction cycles”, the authors mention that the affinity of the branch reaction to the intermediate metabolite of the cycle it consumes must be lower than the affinity of the corresponding recycling reaction of the cycle. Is this statement really correct, given that it is based on either a unscaled sensitivity (dv/ds and not dlnv/dlns) or a saturation degree, which generally also depend on the expression levels of the enzymes, i.e. their Vmaxs?

6) The theoretical predictions could and should be tested more strictly. Although the experimental results that are discussed are consistent with the predictions from the mathematical analysis, they could have alternative explanations. Low saturation of enzymes with their substrates is not uncommon in metabolic networks, and a number of hypotheses have been put forward to explain the apparently wasteful investment in high enzyme concentrations. See for instance refs. [1-6], not to mention some from this manuscript's authors. In fact, while according to the theory a single branching enzyme being less saturated than the cycle enzyme consuming the same substrate would warrant stability, the experimental data point to most of the branching enzymes being less saturated than the cycle enzymes, which makes the concern about alternative explanations pertinent. The authors may thus wish to discuss these potential alternative explanations, and explain why they eventually don't apply. As a negative control, the authors may wish to examine if enzymes branching from non-autocatalytic cycles tend to be more saturated with substrates from the cycle than those branching from autocatalytic cycles.

7) If lower expression of the enzymes catalyzing reactions that branch out from autocatalytic cycles leads to higher production rates of metabolic precursors, one would expect a negative correlation between the cellular abundance of some of these enzymes and growth rate. This prediction should be relatively straightforward to test based on published quantitative proteomics datasets. Again, a comparison with enzymes branching from non-autocatalytic cycles could be made as control.

8) An examination of the regulation of the branching enzymes may also provide additional support for the theory. E.g., these enzymes may be subject to competitive inhibition (to increase the apparent KM) under conditions where there is a higher demand for the cycle's intermediates.

9) It would also be very useful to have experimental data showing that forcing a lower activity or higher saturation of key branching enzymes would cause instability as predicted. However, this may require dedicated experiments, and should not be viewed as a necessary condition for acceptance.

10) While most of the paper seemed clearly written and unambiguous, several statements in the Introduction are imprecise or unclear. (a) One confusing aspect is the ambiguity between collectively autocatalytic systems and autocatalysis of their components. The component of a system that is collectively autocatalytic is in general not autocatalytic. Thus the statement in the second sentence of the Introduction is inaccurate. In fact, the authors themselves contradict that statement in paragraph two of the Introduction. (b) The authors mention that "autocatalytic systems have an inherent potential to be unstable as their operation changes the amount of their components". This last part of the sentence could be true for many other non-autocatalytic dynamical systems, so it is not clear that this statement is justified. (c)The concept of a system being catalytic "with respect to something" is not clearly defined (and, perhaps, unnecessary for clarifying the concept of a purely metabolic autocatalytic cycle); (d) It is not clear how the statement at the end of paragraph two ("Therefore….") follows as a logical consequence of the prior statement.

11) In the interest of reproducibility, the authors should provide supplementary material tables containing the data used for deriving Figure 6, as well as a description of how exactly the data was processed to generate the final flux estimates.

References

[1]. Weiss, S.L., Lee, E.A. & Diamond, J. (1998) Evolutionary matches of enzyme and transporter capacities to dietary substrate loads in the intestinal brush border. Proceedings Of The National Academy Of Sciences Of The United States Of America, 95, 2117-2121.

[2]. Suarez, R.K., Staples, J.F., Lighton, J.R.B. & West, T.G. (1997) Relationships between enzymatic flux capacities and metabolic flux rates: Nonequilibrium reactions in muscle glycolysis. Proceedings Of The National Academy Of Sciences Of The United States Of America, 94, 7065-7069.

[3]. Staples, J.F. & Suarez, R.K. (1997) Honeybee flight muscle phosphoglucose isomerase: Matching enzyme capacities to flux requirements at a near-equilibrium reaction. Journal of Experimental Biology, 200, 1247-1254

[4]. Salvador, A. & Savageau, M.A. (2003) Quantitative evolutionary design of glucose 6-phosphate dehydrogenase expression in human erythrocytes. Proceedings Of The National Academy Of Sciences Of The United States Of America, 100, 14463-14468

[5]. Salvador, A. & Savageau, M.A. (2006) Evolution of enzymes in a series is driven by dissimilar functional demands. Proceedings Of The National Academy Of Sciences Of The United States Of America, 103, 2226-2231

[6]. Eanes, W.F., Merritt, T.J.S., Flowers, J.M., Kumagai, S., Sezgin, E. & Zhu, C.T. (2006) Flux control and excess capacity in the enzymes of glycolysis and their relationship to flight metabolism in *Drosophila melanogaster*. Proceedings Of The National Academy Of Sciences Of The United States Of America, 103, 19413-19418

[7]. Tibor Gánti (2003) "The principles of life" Oxford, Oxford University Press

[8]. Semenov, S.N., Kraft, L.J., Ainla, A., Zhao, M., Baghbanzadeh, M., Campbell, V.E., Kang, K., Fox, J.M. & Whitesides, G.M. (2016) Autocatalytic, bistable, oscillatory networks of biologically relevant organic reactions. Nature, 537, 656-660.

---

## [Author Response]

*Essential revisions:*

*1) In the equation for "fa", there is no mention of the bisubstrate nature of the reaction. It is understood that the authors consider A constant, which seems a reasonable assumption to start with (though it may be interesting to entertain simple alternatives, e.g. as a constant influx of A – see also next point). However, it is not completely obvious that, even if the kinetic parameters relative to A in the M.M. equations are incorporated into the constants of the equation as shown, the bisubstrate nature of the reaction would not affect the generality of the conclusions drawn throughout the paper.*

We thank the reviewers for this (and the following) important points. The reviewers correctly point at an important gap in the paper regarding the bisubstrate nature of the autocatalytic reaction.

We updated the manuscript (subsection “Integrating the bisubstrate nature of the autocatalytic reaction into the simple model”) to include, beyond the uni-substrate case, also four cases of bisubstrate reactions: substituted enzyme (Ping-Pong) scheme, random binding order, ordered binding, with the assimilated metabolite binding first, and ordered binding with the internal metabolite binding first (Materials and methods section "Connecting bisubstrate reaction kinetic constants with simple Michaelis-Menten constants"). We show that in all four cases, keeping A constant results in a M.M. like equation, with kinetic parameters relative to A, maintaining the main result of the paper. In three cases a critical amount for A exists, below which a positive stable steady state cannot be attained (Materials and methods section "Constraints on concentration of assimilated metabolite and kinetic constants of bisubstrate reactions").

In the first three cases of bisubstrate binding, bounds on the affinity of the bisubstrate enzyme to the internal metabolite X, exist (compare, e.g. to equation 5), supporting the conclusions drawn throughout the paper. In the last bisubstrate case (ordered binding with the internal metabolite binding first) the expression levels of the autocatalytic and branch enzymes can be modified to achieve positive stable steady state for any set of affinity parameters.

We therefore conclude that the generality of the conclusions drawn throughout the paper hold also with bisubstrate reactions, with a modification in the case of ordered binding with the internal metabolite binding first. Even for this case the conclusions apply, but do not impose constraints on the affinity of the bisubstrate reaction.

*2) Both in the simple model, and in the extended ones, the authors consider A to be constant. This can be a convenient assumption for the simple model, but makes conclusions for realistic models less relevant, since consumption of A (lowering its level to a critical amount) can be an important part of the system's dynamics. The authors should either explore this alternative possibility, or provide a justification for why this would not be expected to affect significantly the main conclusions.*

As noted in the previous comment, we find that (in all cases but the Ping-Pong reaction scheme) a critical amount for A exists, below which the system loses the positive stable steady state. Interestingly, under some circumstances, an upper bound on the level of A also exists, but this upper bound can be removed by lowering the expression level of the bisubstrate enzyme. The exact critical level of A depends on the bisubstrate reaction scheme and on the specific kinetic parameters and expression levels of the bisubstrate enzyme and the branch enzyme. Nevertheless, qualitatively, the system’s dynamics is identical in all of the reactions schemes we analyze (Materials and methods section "Dependence of steady state concentration on assimilated metabolite"). The concentration of the intermediate metabolite, X, at the st.st. decreases as the level of A decreases (decreasing both the autocatalytic flux, and the branching flux), and reaches 0 when A hits its critical amount (or 0, in the Ping-Pong scheme). Thus, the effect of draining A does not affect the main conclusions, but does add another observation regarding the system’s dynamics when the assimilated metabolite is depleted. We have updated the text to reflect these observations.

*3) Even though the simple example that is treated in the paper nicely sets the stage and helps the reader in understanding the analytical approach, it is based on irreversible enzyme kinetics only. How do the requirements for proper functioning change when the reactions are catalysed by product-sensitive and reversible kinetics? Does product sensitivity prevent loss of stability? How does the displacement from thermodynamic equilibrium of the enzymes at the branch point influence the stability?*

The reviewers address a biologically important extension to the simple example treated in the paper. We added analysis of the requirements for proper functioning under two cases of product-sensitive, reversible kinetics:

1) When the branch reaction is reversible (subsection “Reversible branch reaction can either be far from equilibrium, resulting in the simple case, or near equilibrium, pushing the stability conditions down the branch pathway”), the system behavior depends on the assumptions made regarding the consumption of the product of the branch reaction, Y. If Y is removed rapidly, the branch reaction operates far from thermodynamic equilibrium and the same requirements as in the simple example hold. If Y is removed slowly, then we show that the requirements from the irreversible case are ‘forwarded’ to the reaction that consumes Y, requiring its maximal flux to be higher than that of the autocatalytic reaction, and its flux for small values of Y to be smaller than the autocatalytic flux for small X (Materials and methods section "Reversible branch reaction analysis").

2) When the autocatalytic reaction is product-sensitive (subsection “Analysis of a Reversible autocatalytic cycle reaction”) we find that the level of X for which the reaction proceeds in the autocatalytic direction is bounded (regardless of the exact bisubstrate reaction mechanism). Therefore, in this case the only requirement for proper functioning is that, for constant A, the initial rate of the autocatalytic reaction for small X is larger than the rate of the branch reaction for the same X, a requirement that can always be satisfied by increasing the expression level of the autocatalytic enzyme. Therefore, in this case, product sensitivity prevents loss of stability.

*4) Allosteric regulation, within or onto, the autocatalytic subnetwork can also improve the robust functioning of the autocatalytic subnetwork. Have the authors analysed this? What did they find? This is particularly relevant in relation to the remark, which suggests that enzyme expression adjustment -- so, in fact rate adjustment -- can always lead to a non-zero, stable steady state. If so, then is it true that allosteric regulation could always lead to a steady state? Could one then make predictions about the kinetic design of autocatalytic cycles based on their allosteric control?*

We thank the reviewers for highlighting this important aspect of metabolic control, overlooked in our work. We find allosteric regulation to be valuable in optimizing native autocatalytic cycles enabling both faster convergence to st.st. and modulation of affinities such that branch enzymes can be more efficient, by having higher affinity, when the corresponding autocatalytic subnetwork is not functioning (Materials methods section 13). We note that, as our work also targets synthetic design and metabolic engineering leading to the introduction of novel autocatalytic cycles, we find allosteric regulation implementation to be more complicated in such contexts and thus less appealing a strategy compared with, for example, the introduction of external input fluxes.

We added two sections to the paper analyzing allosteric regulation. One section (Materials and methods section "Allosteric regulation can improve network performance") describes the allosteric regulation interactions (inhibitions or activations) that are expected to improve the function of autocatalytic cycles. Another section (” Analysis of Allosteric regulation potential for cycle improvement”) compares the predicted interactions with the actual direction (activation or inhibition) found for the two autocatalytic cycles we identify and find to be functional in the central carbon metabolism of native *E. coli* (the PTS using cycle and the glyoxylate cycle). Briefly, for the PTS using cycle we find good agreement between our predicted direction of regulation and the regulation found (11/12 interactions agree with our prediction). For the glyoxylate cycle, the agreement is much weaker (8/13 interactions agree with our prediction). We note that this finding is in agreement with the work of Gerosa et al. (Cell Systems 2015) that suggests that TCA cycle fluxes are mostly regulated by transcription and not reactant levels.

Clearly, allosteric regulation serves many more roles and the metabolic network faces many more challenges than just the support of stable autocatalysis. We do not therefore suggest that proper autocatalytic function is the underlying reason for the interactions that agree with our predictions.

*5) In subsection “Stability analysis for multiple-reaction cycles”, the authors mention that the affinity of the branch reaction to the intermediate metabolite of the cycle it consumes must be lower than the affinity of the corresponding recycling reaction of the cycle. Is this statement really correct, given that it is based on either a unscaled sensitivity (dv/ds and not dlnv/dlns) or a saturation degree, which generally also depend on the expression levels of the enzymes, i.e. their Vmaxs?*

We thank the reviewers for showing us we did not make this delicate point clear. We now address the issue of the connection between the unscaled sensitivities of reactions, their saturation degrees, and their affinities in the Materials and methods, section "Inverse relationship between derivatives, affinities, and saturation levels". According to the lemma we prove, we conclude that if f_a(X*)>f_b(X*) and β>α, then it holds that S_a(X*)>S_b(X*) and K_(M,a)<K_(M,b), which correspond to the statements we make in subsection “Analysis of a Reversible autocatalytic cycle reaction”.

*6) The theoretical predictions could and should be tested more strictly. Although the experimental results that are discussed are consistent with the predictions from the mathematical analysis, they could have alternative explanations. Low saturation of enzymes with their substrates is not uncommon in metabolic networks, and a number of hypotheses have been put forward to explain the apparently wasteful investment in high enzyme concentrations. See for instance refs. [1-6], not to mention some from this manuscript's authors. In fact, while according to the theory a single branching enzyme being less saturated than the cycle enzyme consuming the same substrate would warrant stability, the experimental data point to most of the branching enzymes being less saturated than the cycle enzymes, which makes the concern about alternative explanations pertinent. The authors may thus wish to discuss these potential alternative explanations, and explain why they eventually don't apply. As a negative control, the authors may wish to examine if enzymes branching from non-autocatalytic cycles tend to be more saturated with substrates from the cycle than those branching from autocatalytic cycles.*

We thank the reviewers for this point. We agree that alternative explanations may underlie the lower saturation of branching enzymes from autocatalytic cycles.

We added references to the excellent works the reviewers pointed at.

We address this issue in three independent ways:

1) We show that having multiple unsaturated reactions increases the robustness of the cycle to perturbations, as is suggested in minor point 7 below (Materials and methods section "Multiple unsaturated branch reactions increase convergence speed and dampen oscillations").

2) We compare the saturation levels of the same branch points as those presented in Figure 6 for conditions at which the autocatalytic cycles do not function, but the relevant reactions carry flux. We find that in 5 out of 9 cases the saturation levels are reversed, meaning that in these cases the cycle reactions are less saturated than the branch reactions (subsection “Testing the predictions of the analysis with experimental data on functioning autocatalytic cycles” and [Supplementary-material SD1-data]).

3) We consider branching points from two non-autocatalytic cycles – the TCA cycle and the PP + gluconeogenesis cycle (composed of the PP pathway, coupled with gluconeogenic flux in upper glycolysis). For each such cycle we identify the major branch points (akg consumption from the TCA cycle and r5p consumption from the PP cycle). We find that out of 6 branch cases, in 3 cases the branch reaction is less saturated than the cycle reaction, and in 3 cases, the situation is reversed (subsection “Testing the predictions of the analysis with experimental data on functioning autocatalytic cycles” and [Supplementary-material SD1-data]).

To conclude, we find that the bias towards lower saturation of the branch reaction is specific to cases of functioning autocatalytic cycles and is not present neither for the same pairs of reactions in the absence of autocatalysis, nor in branches out of non autocatalytic cycles. We further provide a mathematical analysis highlighting the advantage of multiple unsaturated branch reactions that enable more robust functioning of the cycle.

*7) If lower expression of the enzymes catalyzing reactions that branch out from autocatalytic cycles leads to higher production rates of metabolic precursors, one would expect a negative correlation between the cellular abundance of some of these enzymes and growth rate. This prediction should be relatively straightforward to test based on published quantitative proteomics datasets. Again, a comparison with enzymes branching from non-autocatalytic cycles could be made as control.*

The reviewers suggest an interesting, additional validation of the findings of the paper. However, we do not find this suggestion to be straightforward to test. In order to be tested, it requires multiple conditions with different growth rates, under all of which the same autocatalytic cycle functions and the expression levels of the branch or cycle reactions differ. Glucose-limited chemostat data may seem to be an adequate candidate, but there are a few caveats for using it:

1) The level of the assimilated metabolite, glucose, changes dramatically between the different chemostat conditions.

2) Different transporters exist and function under different concentrations of glucose, not necessarily following the autocatalytic properties of the PTS.

3) As noted in our response to the points regarding allosteric regulation, it may play a major role in controlling fluxes in the PTS-using autocatalytic cycle, which is the cycle used in growth on glucose.

Indeed, evaluating the correlations for two data sets (from Schmidt et al., used also for the expression levels in our article, and from Peebo et al. Mol. Biosyst. 2015) shows that the branch and corresponding recycling enzymes display various dependencies on the growth rate, mostly even non-uniform (that is not monotonically increasing or monotonically decreasing). We believe a specific experimental system, titrating the expression levels of various enzymes along the cycle while maintaining the level of external metabolite fixed, is probably the best way to investigate this phenomenon, but find it to lie outside the scope of this work.

*8) An examination of the regulation of the branching enzymes may also provide additional support for the theory. E.g., these enzymes may be subject to competitive inhibition (to increase the apparent KM) under conditions where there is a higher demand for the cycle's intermediates.*

We thank the reviewers for highlighting this potential additional support. We address this issue together with addressing point 4 above and find that, in the PTS-using cycle, such evidence indeed exists. In the glyoxylate cycle such regulation is not statistically significant, suggesting other control mechanisms may be at play (subsection “Analysis of Allosteric regulation potential for cycle improvement” and [Supplementary-material SD1-data]).

*9) It would also be very useful to have experimental data showing that forcing a lower activity or higher saturation of key branching enzymes would cause instability as predicted. However, this may require dedicated experiments, and should not be viewed as a necessary condition for acceptance.*

We agree with the suggestion the reviewers make here, and find such a system to also be useful for investigating point 7 above further.

*10) While most of the paper seemed clearly written and unambiguous, several statements in the Introduction are imprecise or unclear. (a) One confusing aspect is the ambiguity between collectively autocatalytic systems and autocatalysis of their components. The component of a system that is collectively autocatalytic is in general not autocatalytic. Thus the statement in the second sentence of the Introduction is inaccurate. In fact, the authors themselves contradict that statement in paragraph two of the Introduction. (b) The authors mention that "autocatalytic systems have an inherent potential to be unstable as their operation changes the amount of their components". This last part of the sentence could be true for many other non-autocatalytic dynamical systems, so it is not clear that this statement is justified. (c)The concept of a system being catalytic "with respect to something" is not clearly defined (and, perhaps, unnecessary for clarifying the concept of a purely metabolic autocatalytic cycle); (d) It is not clear how the statement at the end of paragraph two ("Therefore….") follows as a logical consequence of the prior statement.*

We thank the reviewers for pointing out these confusing points regarding the general introduction to autocatalysis. As we found this part of the Introduction does not contribute to the main theme of the paper, we removed it from the revised version.

*11) In the interest of reproducibility, the authors should provide supplementary material tables containing the data used for deriving Figure 6, as well as a description of how exactly the data was processed to generate the final flux estimates.*

The reviewers correctly point at the missing tables, which we added in the revised version.